# Cigarette smoke induces angiogenic activation in the cancer field through dysregulation of an endothelial microRNA
Asawari Korde[1], Anuradha Ramaswamy[1], Seth Anderson [1], Lei Jin[1], Jian-ge Zhang[1], Buqu Hu[1], Walter V. Velasco[2], Lixia Diao [3], Jing Wang [3], Margaret A. Pisani[1], Maor Sauler [1], Daniel J. Boffa[4], Jonathan T. Puchalski[1], Xiting Yan [1], Seyed Javad Moghaddam [2] & Shervin S. Takyar [1] ✉

Cigarette smoke (CS) creates a "cancer field" in the lung that promotes malignant transformation. The molecular changes within this field are not fully characterized. We examined the significance of microRNA-1 (miR-1) downregulation as one of these changes. We found that tumor miR-1 levels in three non-small cell lung cancer cohorts show inverse correlations with the smoking burden. Lung MiR-1 levels follow a spatial gradient, have prognostic significance, and correlate inversely with the molecular markers of injury. In CS-exposed lungs, miR-1 is specifically downregulated in the endothelium. Exposure to CS induces angiogenesis by selectively degrading mature miR-1 via a vascular endothelial growth factor-driven pathway. Applying a multi-step molecular screen, we identified angiogenic genes regulated by miR-1 in the lungs of smokers. Knockdown of one of these genes, Notch homolog protein 3, simulates the anti-angiogenic effects of miR-1. These findings suggest that miR-1 can be used as an indicator of malignant transformation.

## Background

Lung cancer is the leading cause of cancer-related mortality in the United States and worldwide[1,2], and non-small cell lung cancer (NSCLC) comprises over 85% of all types of lung cancer[3,4]. Despite the introduction of novel and impactful interventions in the screening and treatment of NSCLC[5–7], patients diagnosed with this malignancy still face one of the most dismal prognoses among all cancers[8]. This challenging outlook underscores the importance of understanding the basic mechanisms governing the NSCLC initiation and progression.

Cigarette smoke (CS) is the main etiology of NSCLC tumors[9]. Exposure to CS creates a "field of injury" in the lungs and upper airways. This field overlaps with the previously described "cancerization" or "cancer field" where lung cells gain malignant potential and give rise to tumors[10]. The cancer field is delineated by specific molecular alterations in the normal-appearing epithelium that follow spatial and temporal gradients, i.e., become more accentuated near the site of tumors and increase in frequency over time[11,12]. These alterations are not limited to the pulmonary epithelium and encompass a variety of molecular changes in the endothelium and immune cells[11,13]. Angiogenic activation, a cardinal feature of malignancies, is one of these molecular changes[14–16].

MicroRNAs (miRNAs) are small non-coding RNAs that target multiple genes in normal and malignant cells[17] and miRNA dysregulation is among the molecular alterations observed in the NSCLC field[18–20]. MiR-1 downregulation plays a critical role in the angiogenic response and tumor progression, and endothelial-specific overexpression of miR-1 in the lung inhibits KRAS mutant-P53 knockout (KP) tumor formation by more than 90%[21]. Here we show that miR-1 levels have an inverse correlation with the smoking burden and that exposure to CS downregulates miR-1 levels in the murine and human lung endothelium. Using several clinical cohorts, we assess the significance of miR-1 as a marker of field cancerization and examine the mechanism and significance of miR-1 dysregulation in endothelial cells (ECs).

## Results

### MiR-1 has an inverse correlation with the smoking burden

We had previously shown that in lung adenocarcinoma patients tumor miR-1 levels correlate with overall survival[21]. We measured the levels of

[1]Section of Pulmonary, Critical Care and Sleep Medicine, Yale University School of Medicine, New Haven, CT, USA. [2]Department of Pulmonary Medicine, The University of Texas MD Anderson Cancer Center, Houston, TX, USA. [3]Department of Bioinformatics & Computational Biology, The University of Texas MD Anderson Cancer Center, Houston, TX, USA. [4]Department of Surgery, Yale University School of Medicine, New Haven, CT, USA. ✉e-mail: seyedtaghi.takyar@yale.edu

mature (22 nucleotide) miR-1 in a 72-patient NSCLC cohort.(Table S1) Our analysis showed that apart from significant correlations with overall survival (lower miR-1 with shorter survival), histologic subtype (lower miR-1 in adenocarcinoma), and tumor size (lower miR-1 in larger tumors), miR-1 levels had inverse correlations with the history of smoking and cumulative smoke exposure (pack-years of smoking). (Table 1) Consistent with this observation, mean tumor miR-1 levels were significantly lower in smokers vs non-smokers, (Fig. 1A) and also in smokers with more than 30 pack-year history of smoking, compared to those with a lower burden. (Fig. 1B) Similarly, in a second 246-patient NSCLC cohort[22], tumor miR-1 levels were significantly lower in smokers vs non-smokers, (Fig. 1C) and this difference was consistently seen in the subgroup of patients with adenocarcinoma. (Fig. 1D).

To assess the relationship between miR-1 and smoking status in the non-malignant lung tissue, we measured miR-1 levels in samples taken from cytologically normal lungs distant from the tumor site in 21 patients from the Lung Genomic Research Consortium (LGRC) cohort[23]. Similar to tumors, lung tissue miR-1 levels were significantly lower in smokers compared to non-smokers (Fig. 1E). Furthermore, in the samples available from the cytologically normal tumor-adjacent tissues from our cohort, (adjacent tissue samples, AT, Table S2) miR-1 was significantly lower in current smokers compared to former smokers (Fig. 1F). These observations, together with our findings in tumors strongly suggested that CS downregulates miR-1 in the lung and pulmonary tumors.

## Cigarette smoke downregulates miR-1

We tested the effect of CS on miR-1 in both murine and human models of CS exposure. First, we used the murine KRAS-mutant [LSL-Kras[G12D 24]] model, in which genetic activation of the KRAS oncogene in the lung epithelium leads to the formation of adenomatous tumors. Exposure to CS (two months) in this model led to miR-1 downregulation in the tumor-bearing lungs vs controls (Fig. 1G). We found a similar effect in the normal lung. CS exposure decreased miR-1 levels in the lungs of the wild type BL6 mice, confirming that the regulatory effect of CS is not limited to tumors (Fig. 1H). Finally, we tested the effect of CS in human lungs using an in vitro tissue culture model[25]. In these experiments, we used CS extract (CSE) that contains the major carcinogenic compounds of the mainstream smoke[26,27]. As shown in Fig. 1I, CSE had a dose-dependent downregulatory effect on miR-1 in the human lung, confirming our results in the murine models.

## MiR-1 downregulation is a cancerization change

Since miR-1 downregulation is necessary for tumor progression[28], the effect of CS on miR-1 suggested that miR-1 downregulation is a cancerization change. One of the characteristics of cancerization changes is that they follow a spatial gradient in the lung[12,29]. We measured tissue miR-1 levels in a cohort of current smokers in whom both adjacent and distant (contralateral

lung) tissues were sampled by bronchoscopy. (Table S3, Fig. 2A) We found that miR-1 levels were lowest in the tumors (T), comparatively higher in the adjacent tissues (AT), and highest in distant tissues (DT), thus following a spatial gradient. Another characteristic of cancerization changes is their prognostic significance[30]. We performed a Kaplan-Meyer analysis on the T (Table S1) and AT samples (Table S2) in our cohort. These analyses showed that similar to T levels, miR-1 levels in AT samples were predictive of shorter overall survival (Fig. 2B, C).

## MiR-1 levels correlate with the molecular changes of the field of injury

CS exposure induces a "field of injury" in the lungs and airways that overlaps with the cancerization field. Increased expression of the mediators of PI3 kinase (PI3K) pathway is one of these changes[31].We thus asked whether there are associations between miR-1 levels and expression of PI3 kinase (PI3K) pathway mediators in the lungs of patients with NSCLC. We measured the levels of a group of PI3K pathway genes that had been shown to be overexpressed in the field of injury. As shown in Fig. 2D–H, in AT samples miR-1 levels had significant inverse correlations with Phosphatidylinositol-4,5-Bisphosphate 3-Kinase Catalytic Subunit Alpha (PI3KCA), Growth Arrest and DNA Damage Inducible Beta (GADD45B), Forkhead box protein O1 (FOXO1), Mechanistic Target Of Rapamycin Kinase (mTOR1), and AKT Serine/Threonine Kinase 2 (AKT2). Since the expression of these molecules demarcates the field of injury, our findings strongly suggest that a low miR-1 level is also a molecular indicator of the field of injury in smokers.

## CS downregulates miR-1 specifically in the endothelial cells

To define the mechanism and significance of the CS effect on miR-1, we first sought to find the site of this regulation. We thus isolated the epithelial (CD326 + , CD45−), endothelial (CD31 + , CD45−), immune (CD45 + ) and double-negative (CD31−,CD45-) cells from the CSE-exposed human lungs and controls (ex-vivo cultured). Measurement of miR-1 in these cell types showed that among all of them, CSE only downregulated miR-1 in the endothelial cells, (Fig. 3A) confirming our previous findings in the lung[28]. To further assess whether miR-1 dysregulation is specific to the ECs or also occurs in tumor cells, we compared miR-1 levels in an NSCLC cell line vs a TEC model. We exposed a human NSCLC cell line (A549 cells) and a commonly used TEC model (EA.hy926 cells[32,33]), to CSE and compared the dose-response curve between them. CSE downregulated miR-1 in a dose-dependent manner in EAhy296 cells (Fig. 3B). In contrast, miR-1 change in A549 cells was in the opposite direction and lead to higher miR-1 levels at the highest dose. (Fig. 3C), suggesting that CSE has a "direct and specific" effect on endothelial miR-1. To validate and simulate our observations in primary ECs, we tested the effect of CS in human umbilical vein endothelial cells, (HUVECs), and lung microvascular endothelial cells (HPMECs). In both cases CSE had a dose-dependent downregulatory effect (Fig. 3D, E), confirming our findings with the cell lines.

As a further validation step, we asked whether miR-1 changes in ECs follow the same gradient as the trend seen in the whole tissue. Since cancerization changes are most accentuated in the tumor site[12,29] we compared miR-1 levels in tumor endothelial cells (TECs) and non-cancerous pulmonary microvascular endothelial cells (HPMECs). (Fig. 3F) These cells were isolated from a smoker with NSCLC and propagated in culture. Similar to the trend seen in the whole tissue, miR-1 was significantly lower in TECs vs HPMECs, i.e. endothelial miR-1 dysregulation was more accentuated at the tumor site.

## CS regulates miR-1 through VEGF

Since miR-1 is a known VEGF-regulated miRNA[34], we asked whether CS effect is mediated through the VEGF pathway. We first checked VEGF expression in our cohort and found that tumor VEGF levels in smokers were significantly higher than non-smokers. (Fig. 4A) In accord with this finding, exposure to CSE (compared to the control) increased VEGF levels in the ex-vivo human lungs. (Fig. 4B) To find the site of VEGF regulation, we measured its levels in the immune, endothelial, and double-negative (CD31 +

**Table 1 | Clinical association of miR-1 levels in NSCLC tumors**

| Variable | Coefficient correlation (r) | p-value |
|---|---|---|
| Age | 0.015 | 0.897 |
| Sex | | 0.652 |
| **Smoking history** [A] | | **0.018** |
| **Smoking (pack year)** | **−0.233** | **0.048** |
| Tumor size | −0.26 | 0.048 |
| Clinical Stage | −0.157 | 0.215 |
| Histological subtype | | 0.02 |
| Survival | 0.273 | 0.028 |

Bold values highlight the significance of correlation of miR-1 with smoking.

$n$ = 72 all variables except smoking history[68], size[57], stage[63], and survival[63]. [A] smoking history compares miR-1 levels in non-smokers versus smokers. Spearman coefficient correlation was used for all correlations except Age (Pearson coefficient correlation) smoking history (Mann-Whitney test) and survival (Log-Rank Mantel-Cox test).

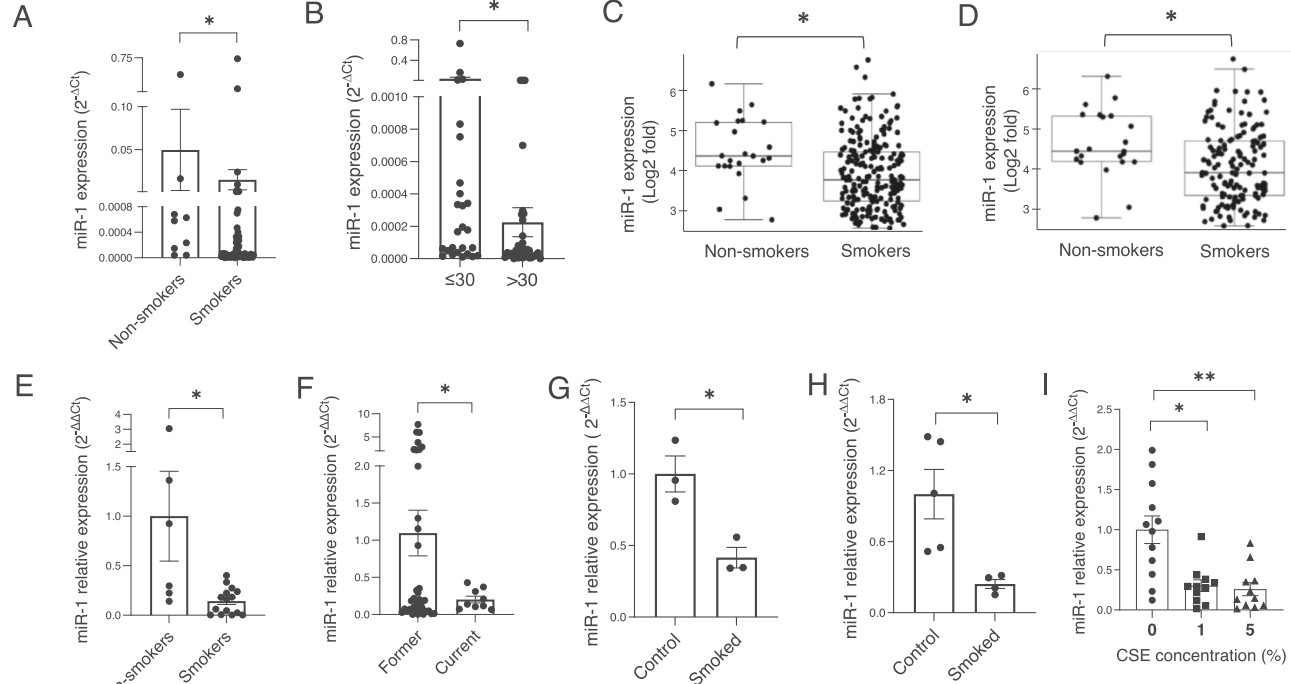

**Fig. 1 | Cigarette smoke and miR-1 levels in NSCLC and non-cancerous lung. A, B** MiR-1 expression (mature miR-1/reference gene, expressed as $2^{-\Delta Ct}$) were measured in tumor samples from NSCLC patients (**A**) MiR-1 in smokers vs non-smokers ($n = 9$ for non-smokers and $n = 63$ for smokers, $p = 0.0186$). **B** MiR-1 in smokers with ≤ 30 vs >30 pack-year smoking history ($n = 27$ for ≤ 30 and $n = 34$ for >30, $p = 0.012$). **C, D** MiR-1 expression in surgically-resected NSCLC tumor samples from patients in the PROSPECT cohort was determined by array analysis (Illumina v3) and expressed as Log2-transformed values. **C** Comparison in all NSCLC patients ($n = 246$, fold $= -1.47$, $*p = 0.004$). **D** Comparison in adenocarcinoma (LUAD) patients ($n = 170$, fold change $= -1.44$, $*p = 0.007$). **E** MiR-1 expression in lung samples from LGRC cohort. Mature miR-1/18S levels were normalized to the mean value of the non-smokers group (control) and expressed as $2^{-\Delta\Delta Ct}$ ($n = 6$ and 15, $*p = 0.0151$). **F** MiR-1 expression in AT tissues from former and current smokers ($n = 38$ and 9 respectively, $*p = 0.0066$). **G** Lungs were harvested from CC-LR (LSL-KrasG12D) mice after exposure to cigarette smoke for 2 months (smoked) or room air (control). MiR-1/reference gene levels were normalized to the mean value from the control group and expressed as $2^{-\Delta\Delta Ct}$. ($n = 3$ per group, $p = 0.02$). **H** Lungs were harvested from mice after 6 months of smoking (smoked) and non-exposed littermates (Control). MiR-1/reference gene levels were measured and expressed as described in (**G**), ($n = 5$ for control and 4 for smoked, $*p = 0.02$). **I** Human lung tissues were cultured ex vivo and exposed to various concentrations of CSE. MiR-1/reference gene levels were measured, normalized to the mean value of the control group(0), and expressed as $2^{-\Delta\Delta Ct}$ ($n = 4$ patients, 3 replicates from each, $*p = 0.0021$ $**p = 0.0014$).

CD45−, CD45 + , and CD45- CD31-, respectively) cells from the CSE-exposed lungs. VEGF levels were only increased in ECs. (Fig. 4C) This finding suggests that, rather than causing the secretion of VEGF from epithelium or immune cells, CS directly regulates the "autocrine VEGF loop"[35,36] and induces the expression of VEGF in the endothelium. We validated this finding using two methods, a dose-response assay and a kinetic experiment. In the dose-response assays, we exposed HUVECs or EAhy926 cells to various doses of CSE and found a dose-dependent increase of VEGF expression in both cell types (Fig. 4D, E). We then selected the highest dose from those experiments (10% CSE) and checked the kinetics of VEGF protein induction after CSE exposure. (Fig. 4F) This experiment showed a significant increase of endothelial VEGF protein ( ~ twice its baseline levels) eight hours after CSE exposure.

The next question to address was whether this autocrine VEGF loop is responsible for miR-1 downregulation. Since autocrine VEGF mediates its effects through VEGF receptor 2 (VEGFR2)[35,36], we tested the effect of VEGFR2 blockade on the CS-miR-1 axis. As shown in Fig. 4G, treatment with SU4156, a commonly used VEGFR2 blocker, abolished the CSE-induced miR-1 downregulation, confirming that VEGF signaling is necessary for the CS effect. Following this line of inquiry, we next asked which signaling pathway downstream of VEGF mediates this regulation. We tested the contribution of four well-known VEGF-downstream pathways[37] by using specific blockers for each. (Fig. 4H) In these experiments we used murine lung microvascular endothelial cells to simulate the conditions in the cancer field. Interestingly, blockade of the PI3 kinase, and specially its downstream mediator, AKT, had the highest efficiency in increasing the

miR-1 levels. AKT blockade even raised miR-1 levels beyond the untreated controls, suggesting that miR-1 is under a tonic inhibitory regulation at baseline.

## CS specifically downregulates the mature miRNA through degradation

Another mechanistic aspect of miRNA regulatory mechanisms is the specific biogenesis step that they regulate[38]. As mentioned earlier, we have measured mature miR-1 in all our assays. To determine the specific biogenetic mechanism targeted by CS, we performed a kinetic study and measured the levels of mature miR-1 and its precursors at various time points after CSE exposure in comparison to the VEGF levels. As shown in Fig. 5A, we found that unlike the majority of the known miRNA regulatory mechanisms[38], CSE exposure does not cause a consistent change in the levels of pri-miR-1 or either of the pre-miR-1 sequences (pre-miR-1-1 and pre-miR-1-2), while it causes a significant and progressive decline of the mature miR-1 levels. This change starts within 6–9 h of exposure and reach its nadir (~30 times decrease) within 12 h, when VEGF levels have risen to more than twice of its baseline levels. (Fig. 4F) This finding, apart from conforming the role of VEGF in this regulation, strongly suggested that the main mechanism for the CS-induced miR-1 downregulation is mature miRNA degradation. To directly test this hypothesis, we asked whether CS has a similar effect on exogenous miR-1. We transfected ECs with a (22nt) double-stranded mature miR-1 mimic. This transfection increases the intracellular mature miR-1 to an easily distinguishable level, more than 50,000-fold higher than the native miR-1 levels at baseline. (Fig. 4D) CSE still had a strong

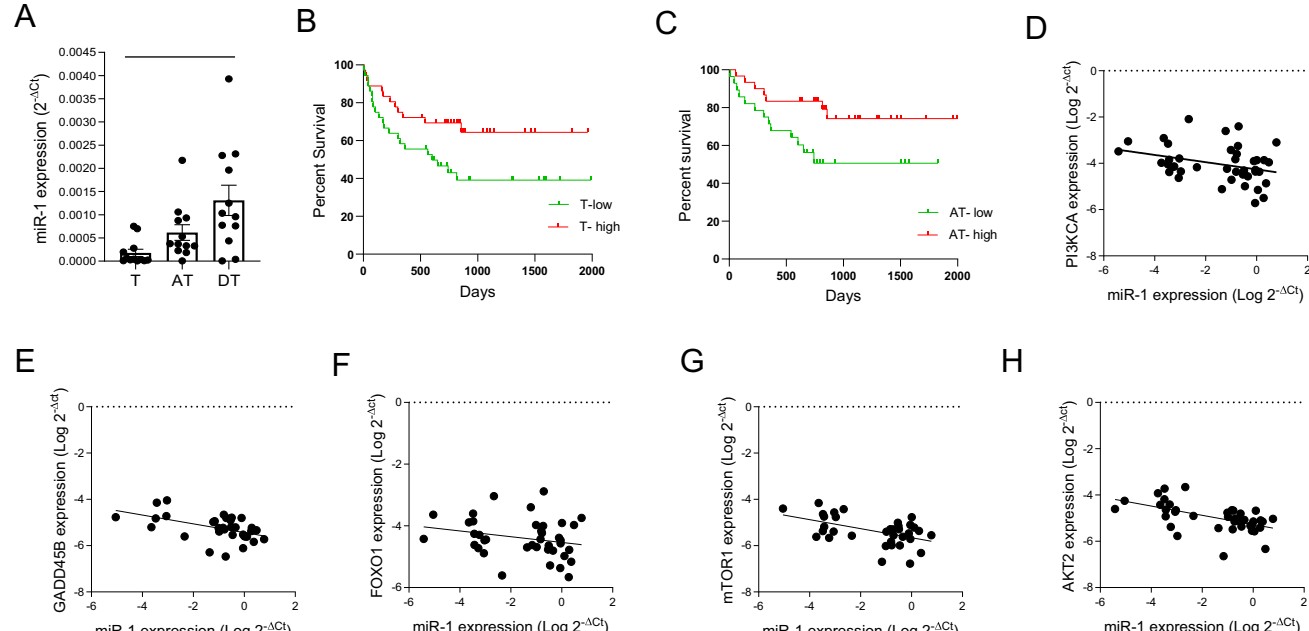

**Fig. 2 | MiR-1 levels in the cancer field. A** MiR-1/ reference gene levels were measured in the tumor (T), adjacent tissue (AT), and distant tissue (DT) samples from smokers and expressed as $2^{-\Delta Ct}$.($n = 12$ patients, $p$-value = 0.0048, one-way ANOVA nonparametric Friedman test). **B, C** T and AT samples were divided into low miR-1( < median) and high miR-1( > median) and the corresponding survivals of the patients in each group were compared using Kaplan Meier analysis (**B**) T-low vs T-high patients ($n = 72$ subjects, $p = 0.0403$, Log-Rank Mantel Cox test) (**C**) AT-low vs AT-high patients ($n = 58$ subjects, $p = 0.0357$, Log-Rank Mantel Cox test). **D–H** The levels of miR-1 and PI3K pathway genes/reference gene were measured in

AT samples (described in table S2) by qRT-PCR and expressed as Log $2^{-\Delta Ct}$. **D** Phosphatidylinositol-4,5-Bisphosphate 3-Kinase Catalytic Subunit Alpha (PI3KCA), (Spearman $r = -0.3491$, $P = 0.0235$, $n = 42$), (**E**) Growth Arrest And DNA Damage Inducible Beta (GADD45B), (Spearman $r = -0.513$, $P = 0.0023$, $n = 33$), (**F**) Forkhead box protein O1 (FOXO1), (Spearman $r = -0.3232$, $P = 0.0419$, $n = 40$) (**G**) Mechanistic Target Of Rapamycin Kinase (mTOR1), (Spearman $r = -0.3499$, $P = 0.0394$, $n = 35$), and (**H**) AKT Serine/Threonine Kinase 2 (AKT2), (Spearman r = $-0.6081$, $P < 0.0001$, $n = 40$).

downregulatory effect and decreased mature miR-1 levels by > 30 times within 6–12 h, confirming that mature miR-1 is directly targeted by a degradation mechanism.

### MiR-1 mediates CS-induced angiogenesis

MiR-1 regulates VEGF-induced EC proliferation and extracellular signal-regulated kinase (Erk) activation[28]. and it is shown that CS has a proan-giogenic effect[39–41]. We thus asked whether miR-1 contributes to the CS-induced angiogenic activation. We first confirmed the angiogenic effects of CS in our experimental setup. CSE exposure increased EC proliferation (Figure S1) and activated Erk (Figure S2). To examine the contribution of miR-1, we first tested the effect of its overexpression. Similar to our previous reports on the effect of miR-1 on VEGF-induced angiogenesis, miR-1 transfection (compared to its scrambled control) reduced EC proliferation (Fig. 6A) and ERK activation (Fig. 6B) in the CSE-exposed ECs, showing that a low miR-1 level is necessary for CS-induced angiogenesis. In complementary experiments, blocking miR-1 with an antagomiR enhanced the proliferative activity of the CSE-exposed cells (Fig. 6C, D), showing that lowering miR-1 is also sufficient to enhance angiogenesis in this context. These two experiments strongly suggest that miR-1 is one of the main mediators of the CS effect on angiogenesis.

### MiR-1 targets a network of genes with known roles in tumor progression

Our findings suggested that miR-1 dysregulation is a field effect that is induced by CS, driven by VEGF, and induces angiogenesis. We used these characteristics to define the specific miR-1 targets in the cancer field. We first performed a multi-step comparative transcriptome analysis on tissue samples from NSCLC patients (6 smokers and 3 non-smokers) and compared the expression of genes between the T and AT samples. Since miRNAs inhibit the expression of their targets, CS downregulates miR-1, and miR-1

levels are lower in tumors vs their adjacent tissues, we sought the genes that were[1] expressed at higher levels in tumors vs adjacent tissues, and[2] were higher in smokers vs non-smokers. This comparison yielded 1695 genes. (Fig. 7A, B) Next, using prediction algorithms we determined that only 486 of these genes had a miR-1 binding site in their 3'UTR (Fig. 7C). We had shown that VEGF drives miR-1 downregulation. So, in the next step we selected the VEGF-driven genes that were previously shown to be expressed in the endothelium [GEO accession numbers GSM162973 and GDS495 and[34]] and ended up with 93 genes (Fig. 7D). Finally, we compared our list with the biochemically-proven miR-1 targets identified in an Argonaute-2 (Ago2) recruitment study on ECs[42], to arrive at the final list of 24 genes with the highest likelihood of being targeted by miR-1 in the cancer field (Fig. 7E, Table S4). Interestingly, 11 of these genes have been previously shown to play a role in tumor progression.

### Biochemical, gene expression, and functional validation of miR-1 targets in endothelial cells

We chose four genes with known or highly likely roles in angiogenesis for further validation: notch homolog protein 3 (NOTCH3), heparan sulfate-glucosamine 3-sulfotransferase 1 (HS3ST1), semaphorin 4B (SEMA4B), and transcription factor AP-4 (TFAP4). Inspection of the predicted binding site sequences assigned to these genes by prediction algorithm (Supplementary fig. S3) revealed that NOTCH3 and HS3ST1 binding sites are canonical high affinity binding site (7-mer-A1, and 7-mer-m8, respectively)[43]) in the 3'UTR. The sites for and TFAP4 and SEMA4B are in the coding region, and thus, considered lower efficiency sites.

To validate the miR-1 targeting and test the efficiency of miR-1-mediated RISC recruitment for each gene we performed an Ago-RNA immunoprecipitation (Ago-RIP) experiment, as described before[44,45]. We measured the levels of each mRNA on Ago-2/input (total mRNA) after transduction with miR-1 overexpression vector vs control. (Fig. 8A). MiR-1

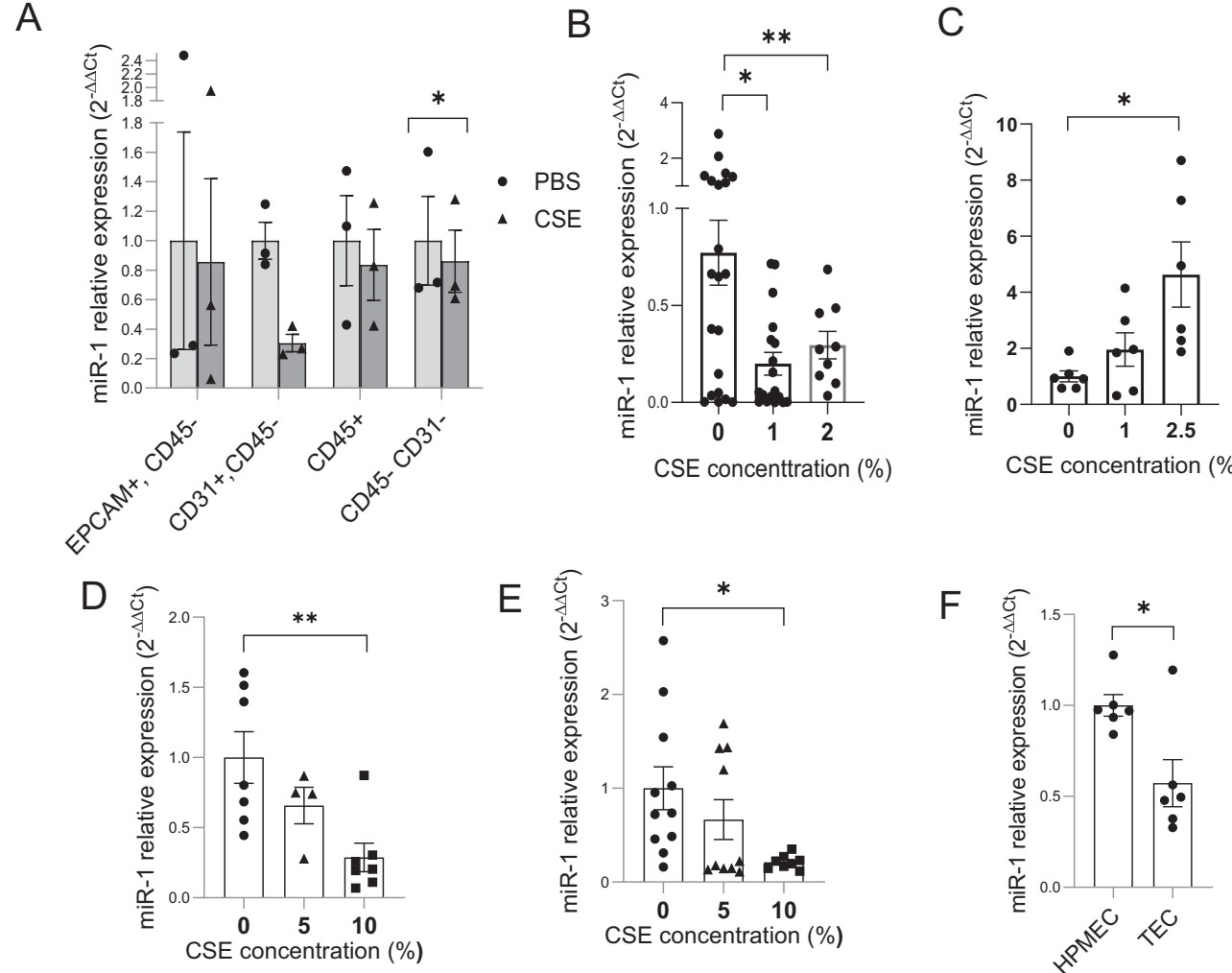

**Fig. 3 | CS downregulates miR-1 specifically in the endothelial cells. A** Human ex-vivo cultured lungs were treated with 1% CSE or vehicle (PBS) for 24 h, and epithelial (CD 45-, EPCAM +), endothelial (CD 45-, CD31 +), immune (CD 45 +), and double-negative (CD45−, CD31−) cells were isolated using magnetic sorting. The graph represents miR-1/reference gene levels in CSE-treated cellular fractions normalized to the levels in PBS groups and expressed as $2^{-\Delta\Delta Ct}$ ($n = 3$ patients, *$p = 0.0176$). **B–E** Cells were treated with increasing concentrations of CSE for 24 h and miR-1/ reference gene levels were measured, normalized to the mean value of '0' CSE concentration group (control), and expressed as $2^{-\Delta\Delta Ct}$ (**B**) EA.hy926 ($n = 21, 4$,

18, and 9 from 4 experiments, *$p = 0.0035$, **$p = 0.00145$) (**C**) A549 cells ($n = 6$/ concentration, from two experiments, *$p = 0.0043$) (**D**) HPMECs ($n = 7$ for 0 and 10% and 4 for 5%, from 2 experiments *$p = 0.0012$) (**E**) HUVECs ($n = 11,10$ and 8, from 3 experiments, *$p = 0.007$) (**F**) Endothelial cells were isolated from non-cancerous lung tissue (HPMECs) and tumor (TEC) from a lung cancer patient and cultured in growth media. miR-1/ reference gene levels were measured in total RNA from the cells, normalized to the mean levels in HPMEC group, and expressed as $2^{-\Delta\Delta Ct}$ ($n = 6$, *$p = 0.041$).

(compared to its control) enriched the levels of NOTCH3 on Ago2 by ~40-folds, compared to ~25-, 10-, and 7- folds for HS3ST1, TFAP4, and SEMA4B, respectively, confirming the predicted high binding affinity of the NOTCH3 site. We next tested the regulatory effects of miR-1 on each gene in an expression assay. (Fig. 8B–E) Exposure of HPMECs to CSE caused significant upregulation of NOTCH3, HS3ST1, and SEMA4B (and a trend toward higher levels for TFAP4). MiR-1 transfection in this context decreased the expression of NOTCH3 5-folds and had a similar regulatory effect (but with lower efficiency) on the other genes. Since both biochemical and gene expression experiments showed the highest regulatory efficiency for NOTCH3, we tested the functional significance of NOTCH3 inhibition in angiogenesis assays. HUVECs were transfected with NOTCH3 siRNA, or control, and exposed to CSE. (Fig. 8F, G) We first evaluated the effect of NOTCH3 knockdown on EC cell number with and without CSE exposure and found that NOTCH3 knockdown, compared to control, decreases the cell number both at baseline and after CSE exposure. To specifically determine the role of NOTCH3 in EC proliferation we next measured the rate of de novo DNA synthesis (BRDU incorporation) after NOTCH3

knockdown and found similar results. These observations show that the effects of NOTCH3 knockdown closely resemble those previously described for miR-1[28], i.e., decreased cell number and de novo DNA synthesis.

### Validation of miR-1 targets in human lung and cancer field

We assessed the translatability of our findings in human lung and NSCLC. We first tested the effect of CSE exposure on these genes in human lung. Consistent with their targeting by miR-1, CSE exposure significantly increased the expression of all of these genes in the ex-vivo cultured human lung tissue. (Fig. 9A–D). Next, we measured the distribution of these genes in our bronchoscopic cohort. As showin in Fig. 9E–H, and in accord with their inhibition by miR-1, the expression of all the genes was the highest at the tumor site (T) and relatively lower in the AT and DT samples.

### Discussion

In this study, we show for the first time that exposure to CS directly and specifically downregulates mature miR-1 in the endothelium of the can-cerization field, leading to angiogenic activation. Our clinical studies show

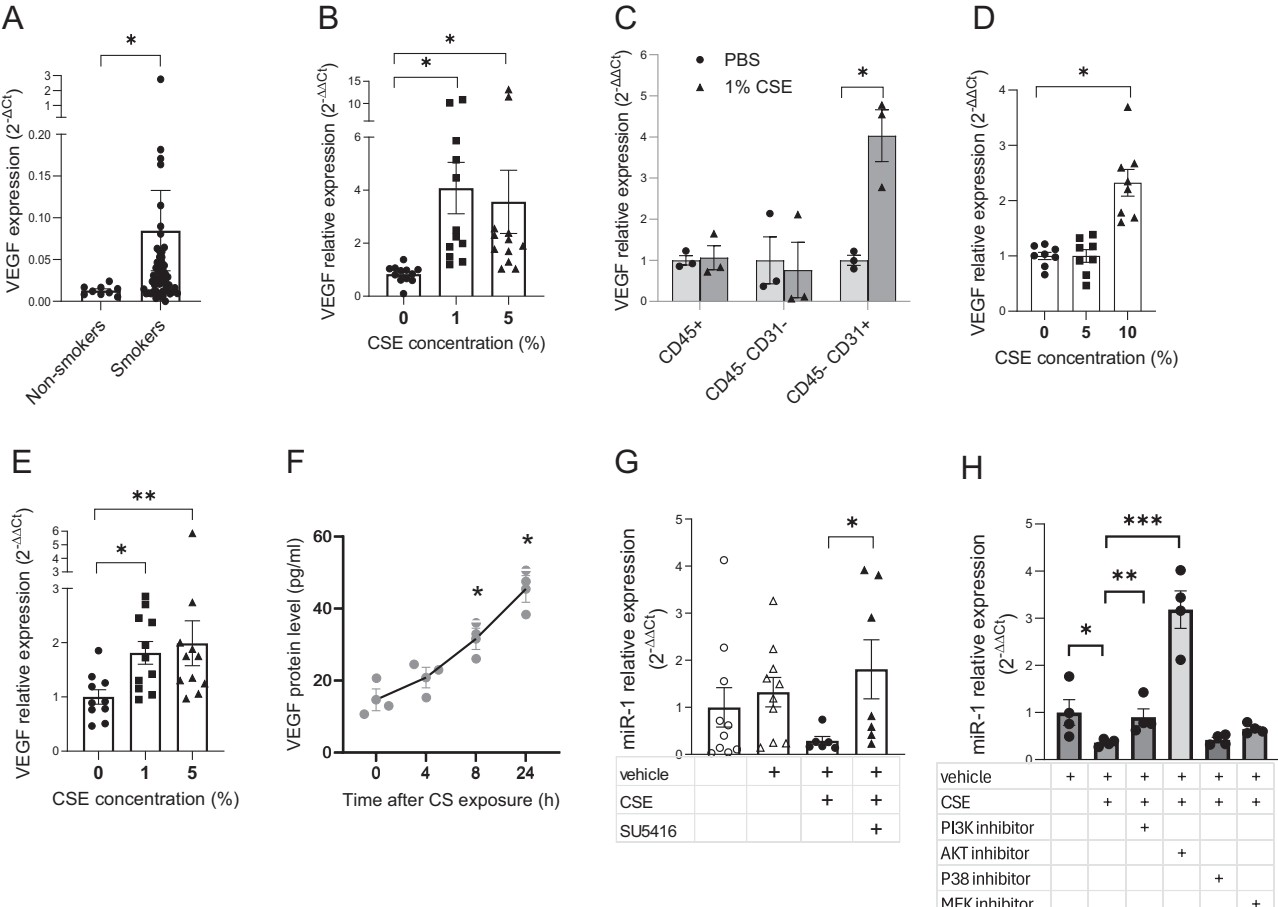

**Fig. 4 | CS downregulates miR-1 through VEGF pathway. A** NSCLC patients were grouped into non-smokers and smokers based on their smoking status and VEGF/reference gene levels were measured in the tumor samples and expressed as $2^{-\Delta Ct}$. ($n = 9$ for non-smokers and $n = 57$ for smokers, $p = 0.0067$). **B** Human lung tissues were cultured ex-vivo and exposed to various concentrations of CSE. VEGF/reference gene levels were measured, normalized to the mean value of the control group(0), and expressed as $2^{-\Delta\Delta Ct}$ ($n = 3$ subjects, 4 replicates in each. *$p < 0.0001$). **C** Human ex-vivo cultured lungs were treated with 1% CSE or vehicle (PBS) for 24 h. Endothelial (CD45− CD31+), immune (CD45+), and double-negative (CD45−, CD31−) cells were isolated using magnetic sorting. The graph represents VEGF/reference gene levels in CSE-treated cellular fractions normalized to the levels in PBS groups and expressed as $2^{-\Delta\Delta Ct}$ ($n = 3$ patients, *$p = 0.0366$). **D, E** Endothelial cells were treated with various concentrations of CSE for 24 h and VEGF/reference gene levels were measured, normalized to the mean value of the control group(0), and expressed as $2^{-\Delta\Delta Ct}$. **D** HUVECs ($n = 8$ in each group, from 2 experiments, *$p = 0.0002$). **E** EAhy926 ($n = 10$ or more from 2 experiments, *$p = 0.0048$, **$p = 0.0062$). **F** HUVECs were exposed to 10% CSE, collected at the time points shown on the X axis, and VEGF protein levels measured by ELISA ($n = 3$/time point, *$p < 0.0$) **(G)** EAhy926 were starved overnight, incubated with and without a VEGFR2 blocker (sUs, and treated with 2% CSE for 24 hours. MiR-1/reference gene levels were measured, normalized to the control group, (vehicle, DMSO) and expressed as $2^{-\Delta\Delta Ct}$ ($n = 6$ or more from 2 experiments, *$p = 0.0221$). **H** Murine lung endothelial cells (MLECs) were treated with blockers, exposed to 5% CSE, and miR-1 was measured and expressed as described in (**G**). ($n = 4$ per group, *$p < 0.03$).

that miR-1 downregulation is a field effect, miR-1 levels in tumors had inverse correlations with the smoking burden, followed a spatial gradient, had prognostic significance, and correlated with the molecular changes typical of the field of injury. Mechanistically, we show that this down-regulation occurs specifically through VEGF-VEGFR2-PI3k-AKT pathway and alters miR-1 levels by inducing the degradation of the mature miRNA. We further illustrate that miR-1 recruits distinct genes to the RISC in the CS-exposed endothelium and that NOTCH3, a high-efficiency target of miR-1, plays a critical role in the pathological angiogenesis induced by CS.

Cancerization changes are defined as molecular abnormalities that are induced by oncogenic agents in the cytologically normal tissue. These changes are most accentuated in tumors and become less prominent further away from the tumor sites, thus following a "gradient distribution" and creating a "field of cancerization" around the tumor[12,46–48]. The most common inducer of malignant transformation in the lung is CS[49,50]. CS exposure has both genotoxic (DNA damage), and non-genotoxic (e.g., cell death, proliferation, and inflammatory) effects on the lung cells[51–55] and thus creates a "field of injury" that overlaps with the cancerization field. The molecular changes of the field of injury mirror the signaling pathways

activated by CS and have been used as prognostic biomarkers for cancer occurrence among smokers[29,31,56]. Increased expression of the mediators of PI3 kinase (PI3K) pathway is one of these changes[31]. We have recently shown that the process of injury downregulates miR-1 in the lung tissue[25]. The association between miR-1 and PI3K pathway mediators strengthens its proposed link to the process of cancerization and suggests that the initial event leading to miR-1 dysregulation in the cancer field may be CS-induced injury.

Even though the initial studies in the field of cancerization focused solely on the epithelial cells, recent findings have implicated tumor microenvironment[57–63]. Endothelium is one of the main constituents of the tumor microenvironment[64–66], and increased angiogenesis has been identified as a marker of field of cancerization[67–71] Furthermore, it is shown that exposure to CS, and specifically nicotine, increases angiogenic cytokines, including VEGF[39,72–74], leading to the survival and proliferation of ECs via the PI3 kinase/Akt[31] and ERK1/2 axes[75,76]. Those observations delineated a path through which CS promotes malignant progression by inducing angiogenesis, and our findings suggest that miR-1 downregulation is a critical step along that path.

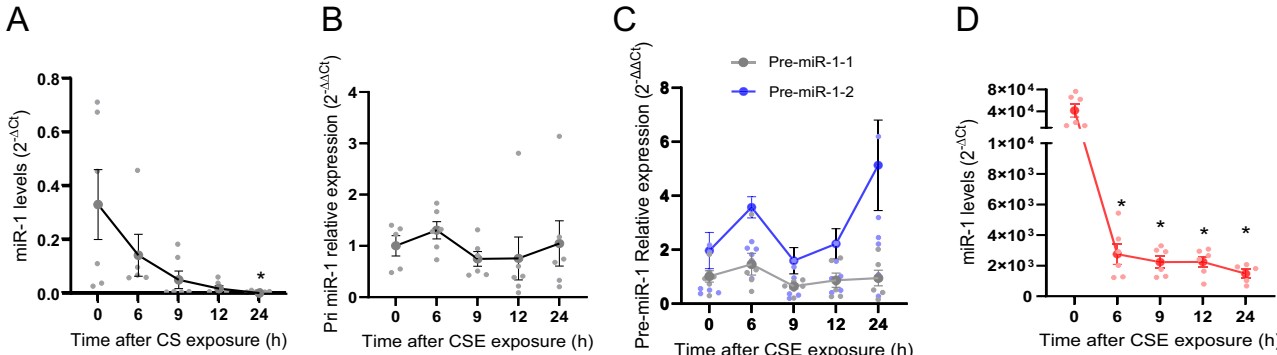

**Fig. 5 | The effect of CS on miR-1 biogenesis. A–C** HUVECs were exposed to increasing concentrations of CSE for 24 h. Cells were collected at the time points illustrated on the X axis and the levels of miR-1/reference gene, (**A**), Pri-miR-1/reference gene (**B**), and pre-miR-1-1 and -2/reference gene (**C**) were measured, normalized to the corresponding mean value at '0' CSE concentration (control), and expressed as $2^{-\Delta\Delta Ct}$. ($n = 5$ /time point, $^*p < 0.01$). **D** HUVECs were transfected with a mature miR-1 mimic (22nt). Twenty-four hours after transfection cells were exposed to CSE and miR-1 levels were measured at various time points as described in (**A**). ($n = 5$ /time point, $^*p < 0.05$).

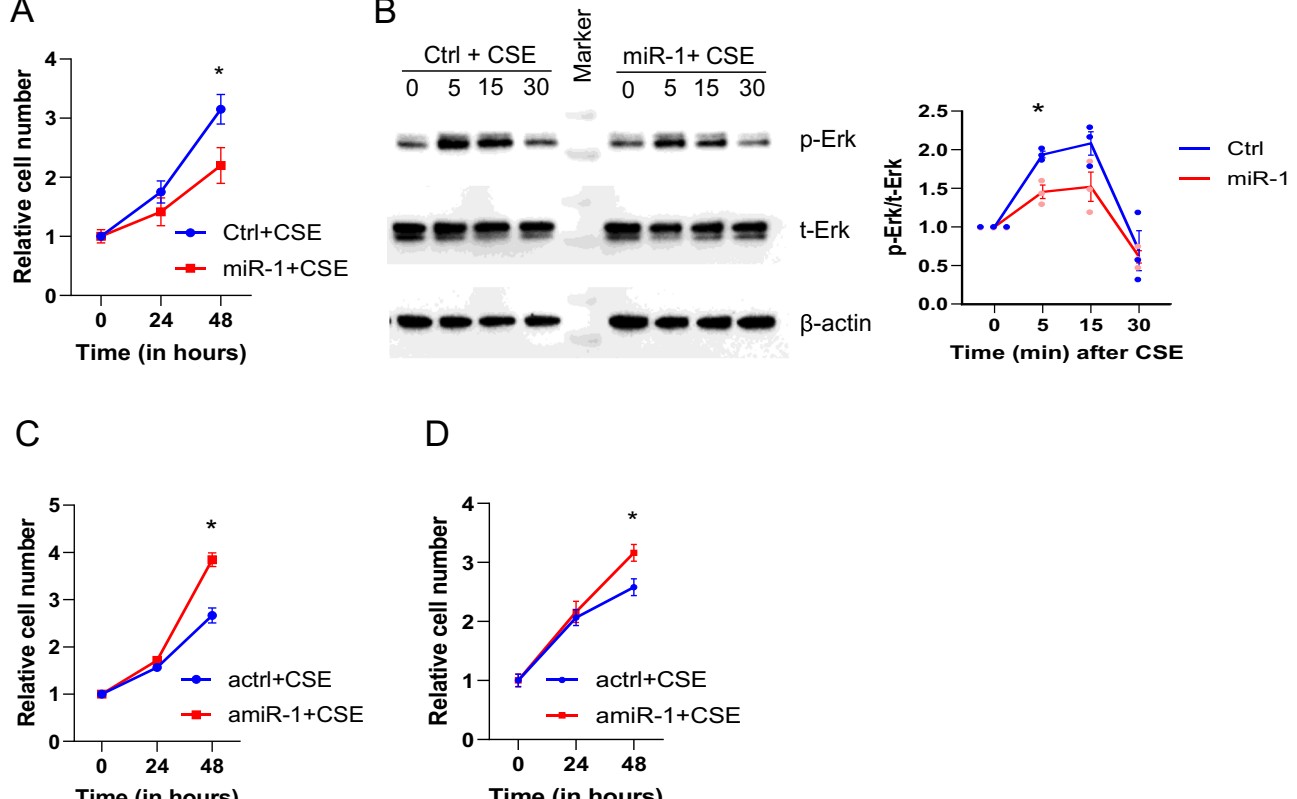

**Fig. 6 | The role of miR-1 in CS-induced angiogenesis. A** EA.hy 926 were transfected with miR-1 or scrambled control RNA (ctrl) and exposed to 1% CSE for the indicated times. Cell numbers were determined and normalized to values at time '0' ($n = 6$ from 2 experiments, $^*p = 0.039$). **B** HUVECs were transfected with miR-1 (or ctrl) treated with CSE and phospho- and total- ERK and beta-actin visualized by Western blot. The top and bottom lines in the marker lane represent 50 and 37 KD bands. **C**, **D** Cells were transfected with antagomiRs (amiR-1 or control RNA, actrl), treated with CSE, and relative cell numbers measured as described in (**A**). **C** EA.hy 926 ($n = 6$, $^*p = 0.0022$) (**D**) HUVECs ($n = 9$ from 2 experiments, $^*p = 0.011$).

Field effects encompass miRNA alterations[19,20,77], and we have focused on an endothelial miRNA in this study. The importance of miR-1 as a tumor suppressor had been reported by various groups. MiR-1 is downregulated in NSCLC tumors[21,78–80], and is one of the four serum miRNAs that are predictive of overall survival in this group[81,82]. Also, tumor miR-1 levels correlated with overall survival[21] and together with PIK3CA, predicted lymph node metastases and 1-year postoperative recurrence[83]. We had shown that miR-1 is downregulated in the tumor endothelium and its downregulation is necessary and sufficient for angiogenesis[28]. In a recent clinical study on

NSCLC patients and cancer-free subjects, serum miR-1 levels had significant inverse correlations with the smoking status in both groups and could distinguish cancer patients from cancer-free subjects[84]. However, the relevance of miR-1 as a field change and the mechanism of its contribution to the cancerization process have never been studied.

Several findings in our work pointed to the significance of miR-1 dysregulation as a cancerization change. First, clinical and experimental observations established the relationship between miR-1 and smoking. In clinical cohorts, miR-1 levels in tumors and non-cancerous lung tissue

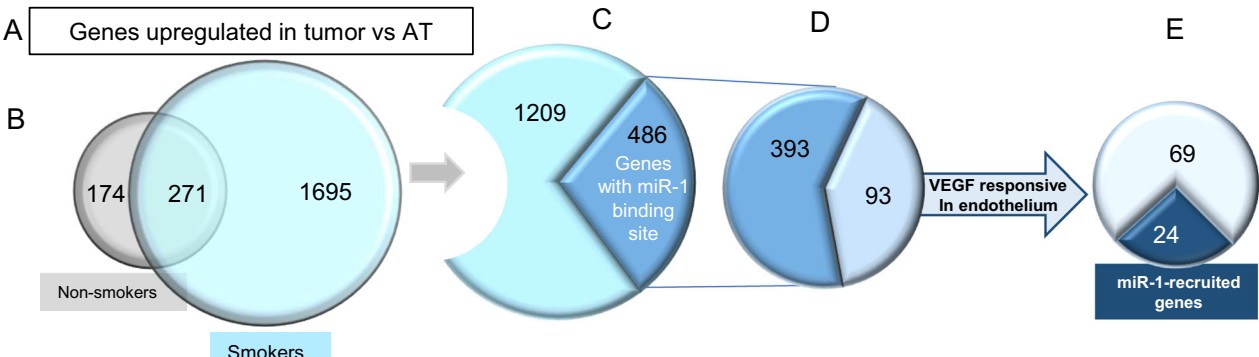

**Fig. 7 | Comparative analysis for miR-1 targets in NSCLC smokers.** We compared gene expression levels between tumor and AT samples from 6 smokers and 3 non-smokers in our cohort. We selected the genes that were (**A**) expressed at a higher level in tumors (vs AT samples) and (**B**) in tumors from smokers vs non-smokers. (**C**) Had a known miR-1 binding site in their 3′UTR (based on three different prediction algorithms) and (**D**) were regulated by VEGF in ECs. **E** We compared the above genes with the list of the genes recruited by miR-1 to the RISC.

correlated inversely with the smoking burden, and in experimental settings, exposure to CS induced miR-1 downregulation in a dose-dependent manner. Second, miR-1 levels in clinical samples fulfilled three main characteristics of cancerization changes; they correlated with the well-established indicators of CS-induced injury, followed a spatial gradient in the airways, and both tumor and adjacent tissue miR-1 levels were predictive of overall survival. Third, miR-1 downregulation has pathophysiological significance. Overexpression of miR-1 in models of NSCLC suppresses tumor growth and propagation[28], and miR-1 downregulation in ECs is necessary and sufficient for angiogenic activation. Finally, miR-1 targets a network of genes with known roles in tumor progression, and those genes also follow a spatial gradient in the airways of smokers.

We found that miR-1 downregulation is a unique intracellular event that unlike the well-described effects of CS on epithelial cells[31], occurs specifically in the endothelium. Using CD31 (PECAM-1), CD326 (EpCAM), and CD45 (Protein Tyrosine Phosphatase Receptor Type C, PTPRC), we isolated 4 different cell types from the CSE-exposed human lung: endothelial, epithelial, immune, and double-negative (CD31-, CD45-) cells. CD31 and CD45 are commonly used as negative markers for purifying pericytes, fibroblasts, and mesenchymal cells[85] and thus, our double-negative fraction contains these cell types. Among all these fractions miR-1 was only downregulated in the ECs. Interestingly, although miR-1 down-regulation has been reported in cancer-associated fibroblasts[86,87], our findings suggest that the change in fibroblasts may not be a direct consequence of CS exposure. The endothelial specificity of this downregulation and its relevance to the cancerization process was corroborated by three further findings: CS induced miR-1 downregulation in a TEC model and in two other primary endothelial cells but not in an NSCLC tumor cell line, and miR-1 levels in TECs were lower than HPMECs, following the same gradient seen in the whole tissue samples. Interestingly, the endothelial-specificity of miR-1 change may explain the reported correlation between serum miR-1 and smoking status, and the accuracy of serum miR-1 in distinguishing NSCLC patients from cancer-free subjects.

The endothelial-specificity of miR-1 dysregulation is also concordant with our findings in the signaling experiments. We found that CS-miR-1 regulation occurs through VEGFR2-PI3K-AKT activation. (Fig. 4B) ECs and TECs have the highest level of VEGFR2 expression among all cell types[88] and VEGFR2 activation in these cells readily leads to the activation of the PI3K-AKT axis[89–91]. Therefore, the endothelial selectivity of miR-1 regulation is most likely due to the involvement of this pathway.

The miRNA biogenetic pathway targeted by CS in this case, is also unique. MiRNAs are processed through several steps, transcription by RNA polymerase II generates primary miRNA (pri-miRNAs) that is cropped by Drosha complex to pre-miRNAs, and finally cleaved by Dicer to produce the mature 22-nucleotide strands for loading onto the RISC[92].Any of these

biogenesis steps can be targeted for regulation. Unlike the majority of miRNA regulations that affect the transcription or processing of the precursors[38], CS did not change the levels of miR-1 precursors. We applied a kinetic approach to determine the mechanism of miR-1 downregulation. In contrast to the progressive decline in the mature miR-1, that starts at 6–9 hours and leads to significantly lower levels between 12 and 24 h, miR-1 precursors (pri- and both pre-miR-1 sequences) fluctuate around the baseline and do not show a significant change.(Fig. 5B, C). These observations strongly suggest that CS does not target transcription, cropping, or dicing of miR-1 precursors. One of the main mechanisms described for regulation of mature miRNAs is degradation[93,94]. We reasoned that if CS causes the degradation of endogenous mature miR-1, it will have the same effect on exogenous miRNA. This approach has been previously used for confirmation of degradation mechanisms[95]. Our experiment with exogenous miR-1 transfection, that distinctively raises the levels of miR-1, showed significant and progressive decline of miR-1 levels after CS exposure, also suggesting that degradation is the main CS-induced mechanism regulating intracellular miR-1 levels. Determination of the type and pathway of miR-1 degradation in this context need further mechanistic experiments.

We found that CS regulates miR-1 through VEGF pathway. This effect is exerted by activation of VEGFR2 and involves the induction of the internal VEGF. Both of these events are critical for endothelial proliferation[36] and thus our observations on the role of CS in angiogenesis are expected. Furthermore, we have previously shown the specific regulatory effect of miR-1 on VEGF-induced EC proliferation and Erk activation[28], and our current findings are consistent with that report. Interestingly, CSE, which contains the main carcinogenic compounds of the mainstream smoke, including nicotine[26,96–99], affects miR-1 similar to the full CS. Since nicotine is a known inducer of the internal VEGF[74,100], these observations raise the possibility that the effects of CSE on miR-1 are due to its nicotine content. The regulatory pathway downstream of VEGF was also similar to the previously described pathway for miR-1 regulation and involved PI3K and AKT[28]. Apart from confirming the role of VEGF in CS-induced miR-1 downregulation, the involvement of the PI3K pathway suggests that the endothelial effects of CS closely resemble its well-described epithelial effects in the field of injury[31]. This similarity may explain the inverse correlations between miR-1 levels and the expression of PI3K genes in the lung. (Fig. 2D–H).

We applied a multi-step screening strategy to delineate a network of endothelial genes downstream of miR-1. We first identified potential miR-1 targets by two comparative analyses in tumors vs their adjacent tissues and in smokers vs non-smokers, and then narrowed down our candidate list by applying two mechanistic constraints[1]: having a predicted miR-1 binding site in the 3′UTR and[2] being driven by VEGF. Finally, we selected the miR-1 targets by considering their recruitment to RISC in a previously published

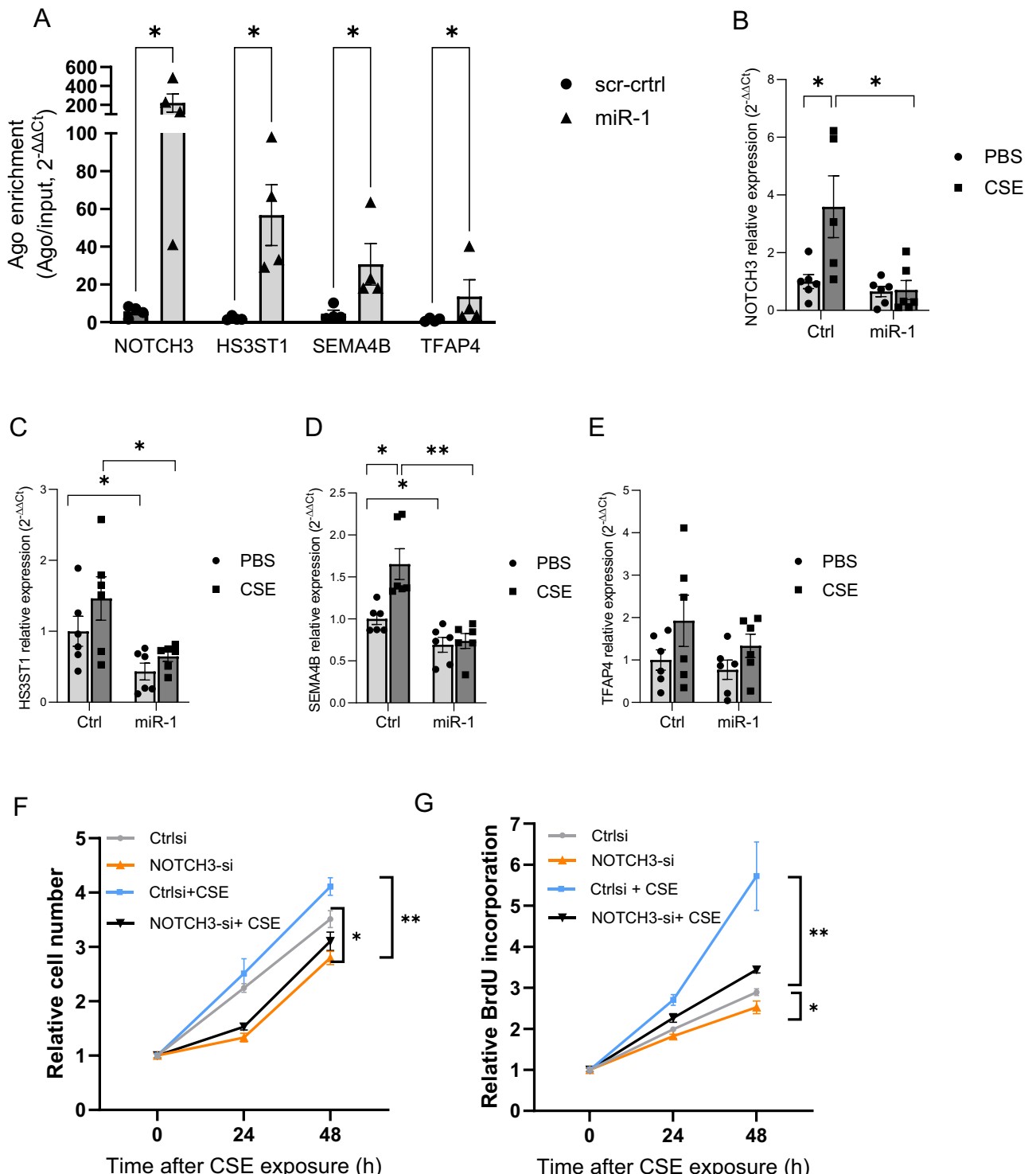

**Fig. 8 | Validation of miR-1 targets in endothelial cells. A** HUVECs were transduced with v-miR-1 (or control vector). Ago-RIP was performed on cell lysates and the levels of each gene in the Ago pull immunoprecipitate/input lysate was measured and expressed as $2^{-\Delta\Delta Ct}$. ($n = 4$, $*p < 0.03$. **B–E** HPMEC were transfected with miR-1 mimic (miR-1) or control RNA (Ctrl) and exposed to 10% CSE for 24 h. mRNA/ reference gene levels were measured, normalized to Ctrl in PBS group, and expressed as $2^{-\Delta\Delta Ct}$. **B** NOTCH3 ($n = 6$, $*p < 0.05$), (**C**) HS3ST1 ($n = 6$, $*p < 0.05$), (**D**) SEMA4B ($n = 6$, $*p < 0.025$,$**p = 0.0026$ (**E**) TFAP4 ($n = 6$, $p = $ NS). **F, G** HUVECs were transfected with NOTCH3 siRNA or scrambled control RNA (ctrl), exposed to 10% CSE for the indicated times. **F** Cell numbers were determined and normalized to values at time '0' ($n = 3$ per group and time, $*p < 0.03$, $**p < 0.02$). **G** De novo DNA synthesis was determined using a BRDU ELISA colorimetric assay. Values were normalized to the baseline (time '0') and presented as relative BRDU incorporation. ($n = 8$ per group and time, $*p < 0.05$, $**p < 0.0002$).

RISC recruitment screen[25]. Many of the genes on this final list had been cited as contributors to tumor progression.

MiR-1 regulates angiogenesis and we have reported a strong correlation between miR-1 and overall survival[28]. Angiogenesis is one of the

hallmarks of cancer[16]and a basic mechanisms of tumor progression and our experimental findings on the CS-miR-1 axis adds to the clinical importance of this pathway. We thus focused our validation experiments on four angiogenic genes that had been shown to have clinical significance.

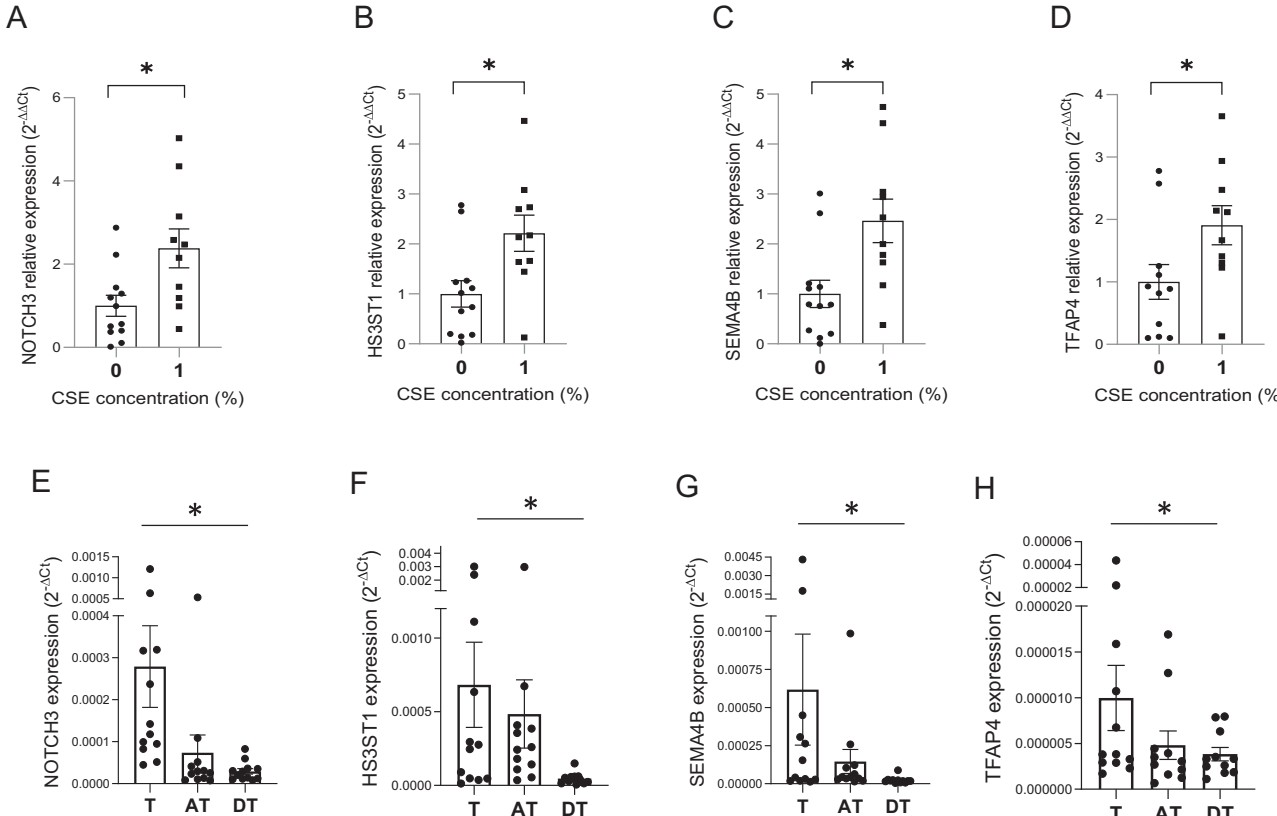

**Fig. 9 | MiR-1 target genes in human lung and cancer field. A–D** Human lung tissues were cultured ex-vivo and exposed to 1% CSE. mRNA/ reference gene levels were measured, normalized to the mean value of the control group(0), and expressed as $2^{-\Delta\Delta Ct}$. **A** NOTCH3 ($n > 10$,*$p = 0.02$) (**B**) HS3ST1 ($n > 10$, *$p = 0.013$) (**C**) SEMA4B ($n > 10$, *$p = 0.0071$) (**D**) TFAP4 ($n > 10$, *$p = 0.024$). **E–H** mRNA/ reference gene levels were measured in the tumor (T), adjacent tissue, (AT), and distant tissue (DT) samples from 12 smokers and expressed as $2^{-\Delta Ct}$. Analysis was done by applying the one-way ANOVA nonparametric Friedman test. **E** NOTCH3 ($n = 12$, $p = 0.0001$), (**F**) HS3ST1 ($n = 12$, $p = 0.0005$). **G** SEMA4B ($n = 12$, $p = 0.0003$), (**H**) TFAP4 ($n = 12$, $p = 0.04$).

(NOTCH3, HS3ST1, SEMA4B, and TFAP4). A direct Ago-RIP experiment confirmed the predicted efficiencies for miR-1 binding to these mRNAs and showed that NOTCH3 is a prime target of miR-1 in CS-exposed ECs. In concordance with these observations, NOTCH3 knockdown simulated the previously described miR-1 effects on EC proliferation.(Fig. 8F, G[28],) NOTCH3 is a critical regulator of developmental and pathological angiogenesis[101]. The levels of NOTCH3 are higher in NSCLC tumors, compared to the adjacent tissues, and correlate with the clinical stage of cancer and metastasis[102,103]. Moreover, it is shown that CS exposure induces NOTCH3 expression and its levels are higher in tumors from smokers vs non-smokers[104]. However, the direct effect of CS on the lung endothelial NOTCH3 has never been described. The similarities between the effects of NOTCH3 knockdown and miR-1 transfection on CSE-induced angiogenesis strongly suggest that NOTCH3 is a main mediator of miR-1 effect in the CSE-exposed cells. Since endothelial NOTCH3 is also induced by hypoxia[101], our observations implicate that NOTCH3 may play a unique role in the "angiogenic switch".

The other identified targets of miR-1 are also known to be involved in angiogenesis and tumor progression. A recent study found that *HS3ST1* levels are higher in NSCLC tumors vs the adjacent tissues and promotes tumor progression[105]. HS3ST1 is one of the sulfation enzymes involved in the post-translational modification of heparan sulfate proteoglycans[106,107]. Endothelial heparan sulfates are integral to angiogenesis[106] and HS3ST1 promotes cell cycle progression by inhibiting spot-type zinc finger protein[105]. Recent reports also support the critical role of *SEMA4B* in NSCLC tumor progression. The expression of SEMA4B in lung adeno-carcinoma strongly correlated with tumor stage and clinical progression[108] SEMA4B belongs to the family of semaphorins, a group of conserved proteins that signal through plexins and/or neuropilins and mediate various

functions in organogenesis, immune regulation, and angiogenesis[109,110]. Several semaphorins (eg Sema 3c, 4D, and 5 A) have been shown to be proangiogenic. The role of Sema4B in angiogenesis has not been clarified yet. TFAP4 is another well-known mediator of tumor progression. TFAP4 is overexpressed in many cancers, including NSCLC[111], and its level is an independent predictor of tumor progression and overall survival[112]. *TFAP4* is a basic helix-loop-helix zinc finger transcription factor that acts as a regulatory hub downstream of c-myc[113], and promotes the invasion and metastasis of cancer cells by regulating the PI3kinase/AKT pathway and increasing the expression of MMP9[114]. Apart from its oncogenic functions in malignant cells, TFAP4 promotes tumor angiogenesis by blocking the inhibitory effect of CCL23[115]. The specific targeting of these genes by miR-1 in CS-exposed endothelial cells implicates that miR-1 dysregulation launches a multi-faceted proangiogenic program by activating heretofore suppressed endothelial genes.

Our studies have a number of shortcomings and potential implications for future work. Our data on the correlation between miR-1 and smoking should be tested in larger clinical cohorts. The concept of cancerization field applies to the initiation of the malignant process. Our previous studies with a triple transgenic mouse model showed the critical role of endothelial miR-1 in tumor formation[28] and our current studies on human and murine lungs and ECs show the specific effect of CS-miR-1 axis on angiogenesis. In our studies, we have tested the effects of CS on three primary cells, HUVECs, human lung microvascular ECs, and human TECs, and one cell line, EAhy.926. These cells mostly mimic angiogenic microvasculature[33,116]. However, CS may affect other vascular beds in the lung and the tissue (e.g., pulmonary veins) with differential effects. Our experiments on the ex-vivo lung, KP mouse model, and a human NSCLC cell line (A549) suggested that miR-1 downregulation occurs specifically in the tumor endothelial cells and

not in the tumor cells. However, since we have not performed extensive testing of human cell lines and tumors, it is possible that a similar down-regulation event may occur in malignant cells. Finally, even though our clinical data, animal studies, the effects of this pathway on endothelial activation, and the targeting of tumorigenesis genes, all strongly suggest that CS-VEGF-miR-1 axis drives tumorigenesis, we have not tested this hypothesis in a transgenic model. Furthermore, our findings suggest that a detailed analysis of the effect of the main components of CSE, including Nicotine, on miR-1 may provide novel insights about the cancerization process.

In summary, we show that CS degrades mature miR-1 in the lung and tumor endothelium through VEGFR2-PI3-AKT axis and this change may contribute to the formation and progression of tumors. The specific molecular characteristics and the functionalities of this miRNA dysregulation make it an ideal candidate for future biomarker development. They also provide the possibility of developing personalized preventive and therapeutic strategies for patients with a high burden of smoking in their history.

## Methods
### Sex as a biological variable
Data from both sexes were included in clinical studies and gender was considered a biological variable in the statistical analysis. In the murine studies, sex was not considered as a biological variable and the studies involved only female mice because of the ease of housing them together. Our findings are expected to be relevant for both males and females.

### Patient characteristics, inclusion criteria, and clinical data collection
Subjects in the Yale NSCLC cohort consisted of patients over the age of 18 who underwent bronchoscopy or surgical resection at Yale New Haven Hospital and based on histopathology were diagnosed with NSCLC. Patients were excluded if they were diagnosed with small cell lung cancer, metastatic cancer from other primary sites, carcinoid or neuroendocrine tumors, benign diagnoses, or if they were referred for repeat biopsies. Surgical patients were recruited during a 1.5-year period from 04/2012 to 11/2013. Bronchoscopy patients were recruited during a 1-year period from 06/2013 to 06/2014. Of the 116 patients initially identified, 38 patients were excluded based on the above criteria, and the remaining 78 patients were included in our study. Forty-four patients were sampled surgically and 34 had bronchoscopic biopsies.

We obtained data from chart reviews for age, sex, smoking history, histological subtype of cancer, size of the tumor (largest tumor dimension in pathology specimens for surgical samples or based on CT Scan images for bronchoscopic patients), and tumor, node, metastasis (TNM) stage of the tumor at the time of the procedure. Overall survival was calculated as time (in days) from the time of the procedure until 8/4/2015 (censoring date). The first sample was collected on 6/7/2012. A chest radiologist reviewed the CT scan images to determine the size of the tumor. We could not reliably determine the TNM stage of the cancer for 8 patients, and the size of the tumor for 14 patients.

### Clinical sample collection and processing
All ethical regulations relevant to human research participants were followed. All human tissue samples were used after receiving written informed consent from patients and after approval by Yale Human Investigation Committee and Institutional Review Board (HIC protocol # 1103008160 for surgical samples and IRB# 0901004619 for bronchoscopic samples). For the PROSPECT cohort, there was no recruitment of human subjects specifically for this study, and details were previously published[117].

Patients were consented for sample collection and measurements before the procedure. Tissue biopsies were de-identified, assigned a code and date, and recorded in the Yale Tumor Registry or Interventional Pulmonology (IP) program database. In surgical sampling, tissues were collected from the tumor and the cancer-free lung tissue within the same lobe. In bronchoscopic sampling, tissues were obtained from the tumor, adjacent cancer-free airway, and contralateral airways. Data from both sexes were included in clinical studies and gender was considered a biological variable in the statistical analysis. Samples were kept in DMEM (Dulbecco Minimal Essential Medium) supplemented with 10% fetal Bovine Serum (FBS) at 4 degrees for less than 4 h before being delivered to the lab for processing. In the lab, each sample was divided into at least two pieces, one for RNA and one for histopathology quantifications. RNA samples were placed in Trizol (Invitrogen) at −80°C and utilized for RNA extraction later. Pathology samples have been kept in O.C.T. medium (Tissue-Tek) at −80°C or formalin-fixed and embedded in paraffin.

### RNA Extraction and miR-1 measurement
RNA extraction was performed using the general Trizol RNA extraction protocol (Life Technologies, Carlsbad, CA, USA). After extraction, RNA concentration and quality of the extracted RNAs were checked using Nanodrop spectrophotometer (Thermo Scientific, Waltham, MA, USA) and Agilent Bioanalyzer, respectively.

Mature miR-1 levels were measured by Taqman quantitative stem-loop RT-qPCR (Life Technologies) or Syber green (#1725271, BioRad) using the regular thermal cycling program 10 mins of initial denaturation at 95 °C, 15 seconds for denaturation at 95 °C per cycle, 30–60 seconds of annealing at 60 °C per cycle and elongation for 30 seconds at 72 °C. Messenger RNAs and miRNA were reverse transcribed using cDNA synthesis kit (BioRad) and miR-1-specific stem-loop or Taqman primers using high-capacity cDNA reverse transcription kit (#4368814, THermoFisher Scientific), respectively, as described in refs. 28,42. For clinical samples, miR-1 and RNU48 (control) levels were measured, and miR-1/RNU48 expression levels were expressed as $2^{-\Delta Ct}$. For other experimental samples, 18S rRNA was used as a control gene for normalization.

### MiR-1 measurement in PROSPECT cohort
The miR-1 expression level was determined by array analysis (Illumina v3) of surgically resected lungs from a total of 246 NSCLC patients consisting of 170 patients with lung adenocarcinomas (LUAD). This cohort was obtained from the Profiling of Resistance patterns and Oncogenic Signaling Pathways in Evaluation of Cancers of the Thorax (PROSPECT) study, developed in 2006 at the University of Texas MD Anderson Cancer Center[22], with known clinical characteristics as we previously described in ref. 118. Expression values were log-(base 2) -transformed. Associations between gene expression and smoking history in all NSCLC and LUAD samples were statistically evaluated using the two-sample t-test. All analyses were performed in the R statistical language and environment (R-project.org; version 3.5.1).

### mRNA expression analysis
Total RNA were reverse transcribed into cDNA using i-script cDNA synthesis kit (BioRad), mRNAs and miR-1 precursors were measured by quantitative real-time PCR (qRT-PCR) using SYBR green (BioRad).18S and human β-actin genes were used as control genes for normalizing the expression of the mRNA in the study. The following primers were used to detect transcripts in human tissues and cells:

NOTCH3: Forward- TGGCGACCTCACTTACGACT, Reverse- CACTGGCAGTTATAGGTGTTGAC;

HS3ST1: Forward-TGGGAGGGAGCATTACAGCCA, Reverse- ACTTTGGGCGACGTGAAATAC;

SEMA4B: Forward- GAGCGGCCATTCCTCAGATTC, Reverse- CACCCACGTACAGGGTCCT;

TFAP4: Forward- GTGCCCACTCAGAAGGTGC, Reverse- GGCTACAGAGCCCTCCTATCA;

Pri-miR-1: Forward- AAACATACTTCTTTATATGCCCA, Reverse-TACATACTTCTTTACATTCCATAGC;

Pre-miR-1-1: Forward- TGGGAAACATACTTCTTTATATG, Reverse-GAGATACATACTTCTTTACATTC;

Pre-miR-1-2: Forward- CCTACTCAGAGTACATACTTC, Reverse-GCCTACCAAAAATACATACTTC;

VEGF-A: Forward- TGCAGATTATGCGGATCAAACC, Reverse- TGCATTCACATTTGTTGTGCTGTAG;

## Preparation of CSE

Mainstream smoke from one 3RF4 research cigarette (University of Kentucky, Lexington, Kentucky, USA) was bubbled through 10 ml of cell culture medium via negative pressure in a fume hood for about 5 min and filtered through a 0.22 µM filter (MilliporeSigma). This smoke-exposed media was considered 100% CSE and mixed with cell culture media in various volume/volume ratios to yield various CSE concentrations[119].

## Murine study

The Institutional Animal Care & Use Committee (IACUC) at Yale University and Animal Care and Use Committee of the University of Texas MD Anderson Cancer Center approved the murine protocol. In the murine study, sex was not considered as a biological variable, and only 6–8-week-old C57BL/6 female mice were used because of the ease of housing them together. However, our findings are expected to be relevant for both males and females. All mice were housed under specific pathogen-free conditions and handled under the guidelines of the IACUC. Mice were monitored daily for evidence of disease or death.

## CC-LR (CCSP$^{Cre}$/LSL-K-ras$^{G12D}$) mouse model

Mice were created by crossing CCSP$^{Cre}$ mouse to LSL-K-ras$^{G12D}$ mouse (at the University of Texas MD Anderson Cancer Center), leading to cell-specific Cre recombination and expression of mutant K-ras only in the club cells of airway epithelium, and subsequent development of lung lesions that progress from atypical adenomatous hyperplasia (AAH) to adenoma and adenocarcinoma as they get older as previously described in ref. 120. Whole-body CS exposure was performed using an InExpose smoke system (SCIREQ Scientific Respiratory Equipment Inc., Montreal, Canada) according to a modified version of a previously described protocol[121] Mice in our study were exposed to CS once daily for two hours, 5 days a week for 2 months. The exposure was conducted by burning 3R4F reference cigarettes (Univ. of Kentucky, TRI). Each cigarette was puffed for 2 s, with a total of 8–10 puffs, each puff in a 1 min period, at a flow rate of 1.05 l/min, to provide a standard puff of 35 cm³. The smoke chambers were monitored daily for total suspended particles and carbon monoxide, with target concentrations of 150 mg/m³ and 500 ppm respectively. For wildtype mice, CS exposure period was extended till 6 months. At the end of the exposure period lungs were harvested for further studies.

## Cell culture

HPMECs were isolated and cultured in Richard Pierce's lab (Yale University) or our lab. Cells were grown in EGM™-2 Endothelial Cell Growth Medium-2 with Bullet Kit (Lonza Cat # CC-3162). HUVECs were purchased from the Yale Vascular Biology and Therapeutics tissue culture core facility and cultured in the M199 (Gibco) medium containing 20% fetal bovine serum (FBS, Heat inactivated, Gibco), 1% ECGS and 1% penicillin and streptomycin (PS). EAhy 926 cells were purchased from ATCC (CRL-2922) and grown in Dulbecco's Modified Eagle's Medium (DMEM) with 10% FBS and 1% PS. Murine lung endothelial cells (MLECs) were isolated and cultured as described previously[21,34]. Briefly, murine lungs were harvested, minced and digested with 0.1% collagenase (Roche, USA) in RPMI 1640 media. After passing through the strainer, cells were pelleted, resuspended in DMEM:F12 medium containing 20% FBS and 1% penicillin and streptomycin and cultured onto gelatin-coated T75 flask. Mouse endothelial cells were isolated on a magnet after treating with biotin-labeled rat anti-mouse CD31 (PECAM-1) antibody (BD PharMingen, CA, USA) and washed with streptavidin magnetic beads (New England Biolabs, MA, USA). Endothelial cells were transfected as described before[28] with miR-1 double-stranded mimic (or control RNA) or antagomiR (with the respective negative controls) at a concentration of 50–100 ηM per reaction using Continuum™ Transfection Reagent (Gemini Bio-products, cat # 400–700) for 24–36 h.

Double-stranded mature miR-1 mimic was prepared as described before[28]. Sense and antisense RNA oligos were synthesized (Integrated DNA Technologies, IDT), hybridized (95 °C for 1' and 37 °C for 30') in 1x siRNA buffer, and diluted with RNase-free water to obtain 50–100 µM concentration. AntagomiRs and the respective control RNA were purchased from Ambion Thermofisher Scientific (cat# AM10617) and diluted as per the manufacturer's instructions.

For proliferation assay, cells were transfected and cell number was counted using a hemocytometer and Trypan blue dye or WST-1 assay (Roche), as described before[28]. Cells were starved for 4 h in serum-free media and then exposed to various concentrations of CSE (concentrate added to the media).

For kinetic analysis (time-course experiment) cells were either transfected with exogenous miR-1 (at 10 pM final concentration) or scrambled control (as described before[34]), exposed to CSE and collected and lysed at various time points.

In blocker experiments, cells were seeded, starved in their respective media with 2%FBS for 6–8 h and treated with blockers for 2 h before CSE exposure. Blockers: SU5416 (Tocris, in DMSO) at 1 uM final concentration, U0126 (MEK1 and 2 inhibitor, Promega, used as ERK inhibitor and solubilized in DMSO) at 15 uM final concentration, SB203580 (P38 MAP Kinase inhibitor, solubilized in DMSO) at 10 uM final concentration, Ly294002 (PI3 kinase inhibitor, Millipore, solubilized in DMSO) at 50 uM final concentration. Cells were lysed 20–24 h after CSE exposure.

## Ex vivo lung culture

Histologically normal lung tissue samples were obtained from patients who underwent surgical resection for lung masses at the Yale Cancer Center under HIC protocol # 1103008160. Samples were collected and transported as mentioned in the 'Clinical sample collection and processing' section. The lung tissue was cut into approximately 3–4 mm size pieces and cultured in M199 (Life Technologies), 20% FBS (Gibco, Life Technologies) at 37 °C[42]. Tissues were then treated with 1 or 5% CSE in growth media and harvested after 24 h for RNA. Tissues were stored in Trizol at −80 °C until RNA extraction.

## Magnetic-Activated Cell Sorting (MACS) for endothelial isolation

Endothelial (CD31+, CD45-), hematopoietic (CD45 + ), and double negative epithelial (CD31−, CD45−) cells were isolated from normal human lung tissues exposed to 1% CSE (or control PBS). Single-cell suspension, microbead labeling, and cell separation using MACS columns were done as described before[42].

## miRISC recruitment analysis and sequencing

HUVEC were transduced with lenti miR-1 (or control lenti, empty vector) followed by Ago2 immunoprecipitation as previously described in ref. 42. RNA was extracted from Ago2-precipitated beads and mRNA analysis was performed as described below in mRNA expression analysis.

## Western analysis

HUVECs were transfected with miR-1 mimic (or control) as described before[28] RNA and exposed to CSE 36 h after transfection. Cells were harvested at designated timepoints for protein extraction and expression of total and phosphor-ERK were measured by western analysis as described before[28].

## Comparative analysis for miR-1 targets in lung cancer

RNA extracted from tumors and AT samples from never-, former-, and current smokers, 3 patients each, were sent for RNA sequencing at Yale Center for Genome Analysis, New Haven, CT, USA. Genes with differential expression above 0.5 Log fold change and p-value less than 0.05 were considered for further analysis. mRNA levels were compared as described in Results. VEGF-responsive genes were identified in the microarray data from mouse lung endothelial cells[34] and from publicly available GEO databases (Accession numbers GSM162973 and GDS495). mRNA containing miR-1

binding sites were identified using miRcode, miRTar base, miRDB, and miRWalk prediction algorithms.

## Statistics and reproducibility

Statistical analyses were performed using GraphPad Prism software version 10.1.2. The normality (Gaussian distribution) of the datasets was tested using D'Agostino-Pearson or Kolmogorov-Smirnov tests. Parametric tests were used for normally distributed data. Normally distributed continuous variables were compared using the student's t-test with Welch correction. Skewed-distributed continuous variables were compared using Mann-Whitney U test. Clinical correlations were analyzed by Pearson's test for parametric data and Spearman's for non-parametric data. Kaplan-Meier curves for the survival of mice were analyzed using Log-rank (Mantel-Cox) test using GraphPad Prism (version 10.1.2). Comparison between the 12-patient cohort was analyzed using one-way ANOVA nonparametric Friedman test. Associations between gene expression and smoking history, separately in all NSCLC and LUAD, were statistically evaluated using the two-sample t-test, and analysis was performed in the R statistical language and environment (R-project.org; version 3.5.1) as previously described[117]. Line bars represent the mean of the group and error bars represent the standard error of mean (SEM). At least 3 biological replicates were used in an experiment and each experiment was repeated at least twice for reproducibility.

## Reporting summary

Further information on research design is available in the Nature Portfolio Reporting Summary linked to this article.

## Data availability

RNA-sequencing data were deposited into the Gene Expression Omnibus Geo database from the National Institutes of Health (NIH) under accession numbers GSE239928 and GSE290190, and are available at the following URL: https://www.ncbi.nlm.nih.gov/geo/query/acc.cgi?acc=GSE239928 and https://www.ncbi.nlm.nih.gov/geo/query/acc.cgi?acc=GSE220190. Values for all data points found in graphs can be found in the 'Supplementary data' file. Additional information or other data are available from the corresponding author upon reasonable request.

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

## Acknowledgements

The authors wish to express their gratitude and appreciation to Dr. P Nana-Sinkham, MD (Virginia Commonwealth University) for reviewing the manuscript, useful critique, and insightful guidance, Dr. Richard Mathay (Yale School of Medicine) for reviewing the manuscript and ongoing support, Dr Jonathan Killam, MD (Dartmouth Hitchcock Medical Center) for expert examination and review of radiologic findings, Dr. Edwin Ostrin, MD, PhD (MD Anderson) for reviewing the manuscript and insightful critique, Dr. Jordan Pober, MD (Yale School of Medicine) for reviewing the manuscript and insightful discussion of the results, and Dr. Mark Godfrey (Hartford health care medical group) for help in collecting and analyzing some of the clinical data. Dr. Maor Sauler has received financial remuneration from Sanofi&Regeneron and Genentech. This work was supported by R00 HL098695 award, NIH, NHLBI (SST), ALA Lung Cancer Discovery award (SST), R01 (R01CA225977), NIH/NCI (SJM) and R01 HL155948, DOD PR211314 (MS).

## Author contributions

A.K., A.R., S.A., L.J., J.G.Z., W.V.V.: experimental; A.K., A.R.: clinical data collection and analysis; A.K., A.R.: writing and formatting the manuscript; A.R., A.K., S.S.T.: clinical and experimental data analysis; X.Y., B.H., L.D. and J.W.: clinical and omics data analysis; M.A.P.: clinical study design, J.T.P. and A.R.: collection and processing of clinical samples, D.B.: collection of clinical samples, M.S. and S.J.M.: Animal model experiment and sample provision, S.J.M.: editing the manuscript; S.S.T.: study conception and design, clinical and experimental data analysis; and writing and editing the manuscript.

## Competing interests

The authors declare no competing interests.
