## [Transparent Peer Review file · Communications Biology]

Cigarette smoke induces angiogenic activation in the cancer field through dysregulation of an endothelial microRNA

Corresponding Author: Dr Seyedtaghi Takyar

Version 0:

Reviewer comments:

Reviewer #1

(Remarks to the Author)

This is an interesting study, which has shown that smokers reduce the expression of miR-1 and low miR-1 is associated with NSCLC progression. Endothelial specific role of miR-1 in modulating NSCLC is interesting. Overall, the authors elegantly designed the study and results support the hypotheses, and manuscript is written well. However, there are following points are below those need to be addressed before it gets published.

- 1) Figure 2; B and D; figure legend needs to be clarified what types of samples were analyzed.
- 2) NSCLC are heterogeneous, other cell types should also be considered. For example, fibroblasts were not included. Comprehensive analysis is needed. What is the expression level of miR-1 in NSCLC cells?
- 3) Figure 4B: Beta actin or Gapdh is also needed in this blot.
- 4) Figure 7: miR-1 regulated genes should be checked in endothelial cell as well.
- 5) Is there any transcription factor (endothelial cell specific) regulates miR-1 expression in endothelium? What is the regulation of miR-1 in endothelium vs NSCLC cells?

Reviewer #2

(Remarks to the Author)

This study focuses on the role of miR-1 in lung cancer and found that cigarette smoking reduces the level of miR-1. Their data further suggest that miR-1 plays a tumor-suppressive role through the downregulation of angiogenesis. It has been reported that cigarette smoke affects many miRNA levels in the lungs. The authors should explain why miRNA-1 is more important than others. microRNAs also suppress protein translation without affecting mRNA level of their target genes. Therefore, this study did not comprehensively analyze the role of miR-1 in lung cancer.

1. The association of miR-1 reduction in lungs with cigarette smoking has been reported in PMID: 32215263. The role of miR-1 in lung cancer has been reported in several studies, such as PMID: 31059033 and PMID: 33305905. The role of miR-1 in suppressing angiogenesis has also been reported in PMID: 28477583. Therefore, the novelty of this current study is low.
2. The level of miR-1 in the tumor and adjacent tumor tissues should be examined by using in situ hybridization assays.
3. In Figure 4, the role of miR-1 in angiogenesis should be proved in some functional assays rather than simply endothelial cell number counting.
4. In Figures 5 and 7, protein levels of VEGF and other potential miR-1-targeting genes should be validated.
5. The provided data was insufficient to conclude the direct targeting of these gene expressions by miR-1.
6. The targeted sequence of the relevant genes by miR-1 should be illustrated.

Reviewer #3

(Remarks to the Author)

The manuscript "Cigarette Smoke Induces Angiogenic Activation in the Cancer Field through Dysregulation of an Endothelial MicroRNA" explores the impact of cigarette smoke (CS) on microRNA-1 (miR-1) regulation in lung endothelial cells and its downstream implications for angiogenesis and tumor progression in the NSCLC cancer field. By examining the

relationship between smoking and miR-1 levels across NSCLC cohorts, murine models, and cell cultures, the authors present evidence for an endothelial-specific downregulation of miR-1. The study positions miR-1 as both a biomarker for the "cancerization field" and a central mediator in CS-driven tumor-promoting angiogenesis.

Major strengths.

1. The study's findings underscore miR-1's potential as a diagnostic and prognostic marker within the "field cancerization" surrounding NSCLC. Given the significant inverse relationship observed between smoking burden and miR-1 expression, this research advances our understanding of CS-induced changes in NSCLC progression, with implications for risk assessment and early detection strategies in smokers.
2. research employs a robust experimental design, utilizing both clinical NSCLC cohorts and studies in cell and animal models. The approach effectively corroborates findings across various systems and provides a multi-faceted understanding of miR-1 regulation. Furthermore, the manuscript's transcriptomic and molecular assays reveal a well-structured investigation into miR-1 targets within the endothelium, lending credence to miR-1's role in angiogenesis.
3. The manuscript clearly demonstrates the relationship between the magnitude of CS exposure and miR-1 downregulation in endothelial cells, supported by data across clinical and experimental models. The inverse correlation between miR-1 levels and smoking intensity is evident not only in NSCLC tumors but also in non-malignant lung tissues, reinforcing the association between CS and endothelial miR-1 dysregulation.

Major Limitations

1. Although the manuscript effectively correlates smoking with miR-1 dysregulation, it does not establish a detailed mechanistic pathway by which CS alters miR-1 levels specifically in endothelial cells. An in-depth exploration of potential signaling mechanisms or intermediate molecular factors would add to the translational significance of these findings, particularly for targeted therapeutic approaches.
2. While the authors identify several miR-1 target genes associated with tumor progression, the functional relationship between these genes and NSCLC pathophysiology is not fully elucidated. Expanding on how miR-1's regulation of these targets directly influences key cancer processes, such as invasion or metastasis in NSCLC, would enhance the impact of the research. A mechanistic assessment of the impact of changes in miR-1 or select targets on "field cancerization" or NSCLC characteristics that would pose a worse prognosis would strengthen the report.
3. The manuscript does not address potential variations in miR-1 regulation across different endothelial cell subtypes within the lung, which could lead to differential impact on angiogenesis. Does inhibition of miR-1 (by CS) in microvascular lung endothelial cells cause upregulation of similar targets as in NSCLC-derived endothelial cells or pulmonary artery endothelial cells. Investigating this heterogeneity could add valuable context and specificity to the study's conclusions regarding miR-1's role in the cancerization field vs. the tumor itself vs. non-malignant detrimental effects of CS on the lung.

Version 1:

Reviewer comments:

Reviewer #1

(Remarks to the Author)

The authors have done a great job in revising the manuscript. However, this reviewer is not convinced that miR-1 is not reduced in the cancer cells using CSE exposure, as the authors used only one cell line, A549. It should be done in multiple NSCLC cell lines. In addition, Figure 6B should be quantitated.

Reviewer #2

(Remarks to the Author)

The authors addressed the comments by this reviewer. The current version would be suitable for publication.

Reviewer #3

(Remarks to the Author)

the authors have responded to 1 of the 3 major critiques with new experiments (critique 1). I think addressing critique 2 with clarifications in the discussion is acceptable, but I was disappointed that they did not at least in part address critique #3 with an experiment testing for of a couple of major mir1 targets in human large pulmonary artery endothelial cells which are available commercially (HUVECs are irrelevant to lung biology). However, the absence of those experiments will not deter from main focus of the report. As minor comment, there is a reference listed as REF(JEM) that was intended to be inserted but was not added via reference manager.

Response to referees

Below is our extended response to each reviewer comment. Our responses are marked by a dash and indent after each comment.

The changes in the text are highlighted in yellow.

Reviewers' comments:

Reviewer #1 (Remarks to the Author):

This is an interesting study, which has shown that smokers reduce the expression of miR-1 and low miR-1 is associated with NSCLC progression. Endothelial specific role of miR-1 in modulating NSCLC is interesting. Overall, the authors elegantly designed the study and results support the hypotheses, and manuscript is written well. However, there are following points are below those need to be addressed before it gets published.

-We appreciate the reviewer's comments and their attention to detail. The answer to each comment is explained after the questions.

1) Figure 2; B and D; figure legend needs to be clarified what types of samples were analyzed.

-The samples analyzed in Figure 2b are the T and AT samples described in tables S1 and S2, as described in the text. We have now clarified this point in the legend as well.

-The samples in Figure 2(D-I) are the AT samples described in table S2, as mentioned in the text. We have added this point to the legend now.

2) NSCLC are heterogeneous, other cell types should also be considered. For example, fibroblasts were not included. Comprehensive analysis is needed. What is the expression level of miR-1 in NSCLC cells?

-We appreciate the reviewer's concern. We have added new data, and modified the Results and Discussion Sections. Our current manuscript is mainly focused on the cancer field that corresponds to the CS-exposed lung. To address the issue of heterogeneity in the lung we isolated and tested epithelial (CD326+ CD45-), endothelial (CD31+ CD45-), immune (CD45+), and double negative (CD45- CD31-) fractions. (new Figure 3A) The double negative fraction contains all other cell types, including pericytes, mesenchymal cells, and fibroblasts. Analysis of all these fractions showed that CS downregulates miR-1 only in the endothelial fraction. Although the regulation of miR-1 in fibroblasts has been reported before (1, 2), it is likely that CS does not directly affects

miR-1 in fibroblasts. We have updated Figure 3A with the data on the 4 cell types, modified the results section, and added this point to the discussion.

The other aspect of heterogeneity is the distinction between the tumor and TME (tumor endothelium in this case). In our previous report on the role of miR-1 in angiogenesis, we fractionated the NSCLC samples from KP transgenic mice into CD31+, CD45- (endothelial) and CD45+ (immune) compartments (Figure 3B of that manuscript,(3) and found that miR-1 downregulation occurs specifically in the endothelial cells. To make this distinction with the CS in this manuscript we performed a new experiment on a human NSCLC cell line (A549) and compared the results with an endothelial cell line (EAhy926) that is commonly used as a TEC model. We have put these new results in Figure 3 and added these findings and its implication to the Results and Discussion Sections, respectively.

3) Figure 4B: Beta actin or Gapdh is also needed in this blot.

-We repeated the experiment several times and managed to visualize the three bands for the phosphorylated and total ERK and beta actin on the same blot. (Figure 6B) The main difficulty with this particular experiment is that the bands for ERK, beta actin, and GAPDH run close to each other on regular blots, and with repeated stripping it becomes difficult to visualize the house keeping gene. Nevertheless, we ran the experiment on a higher density gel and managed to see the three bands. The other point is that while we agree that beta actin or another housekeeping gene is commonly used for loading control, the value of such a control is much less in signaling gels because any loading correction would affect both total and phosphorylated bands and cancel out. This is the reason why we have been able to publish signaling gels without a beta actin control before.(3, 4) Finally, all these results closely resemble our previous reports on the effect of miR-1 on VEGF-induced ERK activation(3) and since we are reporting that CSE works through VEGF, it is almost redundant to repeat all the experiments.

4) Figure 7: miR-1 regulated genes should be checked in endothelial cell as well.

- We appreciate the Reviewer's comment and agree that showing the CS effect on the targets in the endothelial cells is of utmost importance. All the data in the new Figure 8, biochemical validation of the targets, expression studies and functional validation of NOTCH3 and are performed in endothelial cells and we have clarified this point in the titles in the text and the figure legend.

5) Is there any transcription factor (endothelial cell specific) regulates miR-1 expression in endothelium? What is the regulation of miR-1 in endothelium vs NSCLC cells?

-We agree that finding a mechanism for endothelial specificity is critical. Our new data on pre and pri-miR-1 shows that miR-1 is not controlled at the level of transcription, rather the regulation occurs at the level of mature miRNA and through a degradation mechanism. (new Figure 5) Our data in Figure 4 show that this regulation is driven by VEGF through VEGFR2. In our new experiments we have also found that CS exposure specifically regulates miR-1 by activating PI3 kinase-AKT pathway. Since ECs and TECs have the highest level of VEGFR2 expression among all cell types(5) and VEGFR2 activation in these cells readily leads to the activation of the PI3K-AKT axis,(6) the endothelial selectivity of miR-1 regulation is most likely due to the involvement of this pathway. We have mentioned this point in discussion.

Reviewer #2 (Remarks to the Author):

This study focuses on the role of miR-1 in lung cancer and found that cigarette smoking reduces the level of miR-1. Their data further suggest that miR-1 plays a tumor-suppressive role through the downregulation of angiogenesis. It has been reported that cigarette smoke affects many miRNA levels in the lungs. The authors should explain why miRNA-1 is more important than others. microRNAs also suppress protein translation without affecting mRNA level of their target genes. Therefore, this study did not comprehensively analyze the role of miR-1 in lung cancer.

1. The association of miR-1 reduction in lungs with cigarette smoking has been reported in PMID: 32215263. The role of miR-1 in lung cancer has been reported in several studies, such as PMID: 31059033 and PMID: 33305905. The role of miR-1 in suppressing angiogenesis has also been reported in PMID: 28477583. Therefore, the novelty of this current study is low.

-We appreciate the reviewer's concern about novelty. We agree that it is necessary to expand on the novel aspects of our findings and clarify these aspects compared to the previous data. We have clarified that a direct effect of CS on the endothelium in the cancerization field, causing angiogenic activation, has never been reported in the literature (including the references cited by the reviewer, reviewed below) in the first paragraph of the discussion. We have also added three new experimental findings to the manuscript to enhance the novelty of our manuscript. The new experimental findings are as follows. In each case we have modified the results section and added the relevant point to the discussion:

- 1- Evidence for a miR-1 degradation mechanism in the CS-exposed endothelium. We performed a kinetic analysis of the VEGF protein (Figure 4F), the endogenous mature miR-1 and its precursors (Figures 5A-C), and exogenous

(transfected) miR-1 (Figure 5D) and found that both endogenous and exogenous miR-1 levels decline significantly over 6-12 hours after CSE exposure, and this decline is synchronous with a rise in VEGF levels.

- 2- Endothelial signaling pathway downstream of CS: Using specific inhibitors we have now shown that CS specifically activates PI3K-AKT axis downstream of VEGFR2, without significantly altering P38MAPK or ERK pathways. The selective involvement of this pathway in CS-induced miRNA degradation has never been described. We clarified this point in the discussion
- 3- Using an Ago-RIP, expression and angiogenesis assays we now show that NOTCH3 is a main mediator of the CSE effect on angiogenic activation. Although the effect of CS on NOTCH has been described recently,(7) the activation of this pathway in the lung endothelium by CSE, and the role of this pathway in angiogenesis are novel.

The following is a review of the cited articles, the major differences between these reports and our work, and the changes in the manuscripts including addition of the relevant references:

PMID: 32215263 describes the findings on plasma samples from 47 NSCLC patients and 41 cancer-free subjects and analyzes the diagnostic properties of plasma miR-1 for NSCLC diagnosis. It also shows an association between plasma miR-1 levels and smoking status. This article has no description/studies on the origin, mechanism, or significance of the findings in serum miR-1. However, even though the size of this cohort is limited, and no mechanistic investigations has been performed, the observations in this clinical article may be relevant clinically and are most probably explained by our findings. We have added this reference and this point to our discussion.

PMID: 31059033 is a meta analysis of clinical studies, evaluating miR-1 as a biomarker of lung squamous cell carcinoma in publicly available databases, and a bioinformatic analysis to predict potential cellular pathways regulated by miR-1, with no experimental data to validate any of the targets or reference to the site of miR-1 downregulation in the tumor. We have added this reference as another citation for the clinical significance of miR-1.

PMID: 33305905 focused on the change in miR-1 level after the occurrence of resistance to epidermal growth factor receptor tyrosine kinase inhibitor (EGFR-TKI) and then performed experiments with miR-1 transfection in one tumor cell line. These findings on the change in miR-1 after EGFR-TKI resistance in tumor cells are not relevant to our findings and show a different pathway that is not related to CS exposure either. Our new experiment with CS on A549 (Figure 3C) and previous report on the origin of miR-1 dysregulation in tumors and tumor models (3) confirm the endothelial-specificity of miR-1 change.

PMID: 28477583 is a tissue engineering article showing that hypoxic conditioning of myoblasts leads to downregulation of miR-1, miR-206, and Angiopoietin-2, and upregulation of VEGF in those cells. These myoblasts could then cause increased angiogenic activity in a vascular pedicle in a severe combined immunodeficiency mice. However, in a transfection experiment they show that miR-206 controls VEGF levels in myoblasts. It is not clear to us if the findings in this manuscript are related to our reports on the direct effect of cigarette smoke on endothelial cells.

2. The level of miR-1 in the tumor and adjacent tumor tissues should be examined by using in situ hybridization assays.

-In situ hybridization is occasionally used as a confirmatory experiment to show overexpression of miRNA in tissue samples. There are several technical difficulties with the application of this method in this case. MiR-1 is downregulated only in the endothelium and therefore harder to differentiate in the tissue. Also, the regulatory mechanism is specifically affecting the mature miRNA and hybridization probes do not readily differentiate the mature from the precursors. We have unsuccessfully tried tissue hybridization previously and believe that optimization of this method is out of the scope of this manuscript.

3. In Figure 4, the role of miR-1 in angiogenesis should be proved in some functional assays rather than simply endothelial cell number counting.

-We appreciate the reviewer's concern about the functional significance of our findings. We have previously published an extensive report of the effect of miR-1 on VEGF-induced angiogenesis, addressing every aspect of angiogenesis (signaling, cell proliferation cell death, differentiation, and migration). Our current report is showing that miR-1 affects CSE-induced angiogenesis, and that CSE effect is mediated through VEGF. Therefore, we have only shown the effects on signaling and cell proliferation and think that repeating all the miR-1 experiments would be redundant. However, we have clarified this point in the results and added a reference to our previous findings on the specific anti-proliferative effects of miR-1 In the discussion. Also, in our new experiment on NOTCH3, a target of miR-1, we show the specific targeting of proliferation using a BRDU incorporation assay. (Figure 8G)

4. In Figures 5 and 7, protein levels of VEGF and other potential miR-1-targeting genes should be validated.

-We agree that that showing the effects of CS on VEGF protein levels will add to the experimental vigor of the paper. We have measured VEGF levels in CS-exposed

ECs in a new figure (new Figure 4F) that kinetically corresponds to both figures 5 and 7. Showing the effect of CS on the protein levels of the target genes will need extensive optimization of various antibodies and in our opinion, outside the scope of this manuscript.

5. The provided data was insufficient to conclude the direct targeting of these gene expressions by miR-1.

-The current targets were not selected solely based on transcriptome analysis. The last step of our workflow consisted of identifying target genes based on a previously published Ago-IP screen, which is an accepted method of miRNA targeting.(8) However, in order to address the reviewer's comment, we repeated an Ago-RIP experiment on miR-1 vs control-transfected ECs and preformed gene specific qPCR assays for the selected genes on the input and Ago-IP lysates. (Figure 8A)

6. The targeted sequence of the relevant genes by miR-1 should be illustrated.

-We have illustrated the position and sequence of the miR-1 binding site in each mRNA in Supplementary Figure 3 and referred to them in the text.

Reviewer #3 (Remarks to the Author):

The manuscript "Cigarette Smoke Induces Angiogenic Activation in the Cancer Field through Dysregulation of an Endothelial MicroRNA" explores the impact of cigarette smoke (CS) on microRNA-1 (miR-1) regulation in lung endothelial cells and its downstream implications for angiogenesis and tumor progression in the NSCLC cancer field. By examining the relationship between smoking and miR-1 levels across NSCLC cohorts, murine models, and cell cultures, the authors present evidence for an endothelial-specific downregulation of miR-1. The study positions miR-1 as both a biomarker for the "cancerization field" and a central mediator in CS-driven tumor-promoting angiogenesis.

Major strengths.

1. The study's findings underscore miR-1's potential as a diagnostic and prognostic marker within the "field cancerization" surrounding NSCLC. Given the significant inverse

relationship observed between smoking burden and miR-1 expression, this research advances our understanding of CS-induced changes in NSCLC progression, with implications for risk assessment and early detection strategies in smokers.

2. research employs a robust experimental design, utilizing both clinical NSCLC cohorts and studies in cell and animal models. The approach effectively corroborates findings across various systems and provides a multi-faceted understanding of miR-1 regulation. Furthermore, the manuscript's transcriptomic and molecular assays reveal a well-structured investigation into miR-1 targets within the endothelium, lending credence to miR-1's role in angiogenesis.

3. The manuscript clearly demonstrates the relationship between the magnitude of CS exposure and miR-1 downregulation in endothelial cells, supported by data across clinical and experimental models. The inverse correlation between miR-1 levels and smoking intensity is evident not only in NSCLC tumors but also in non-malignant lung tissues, reinforcing the association between CS and endothelial miR-1 dysregulation.

Major Limitations

1. Although the manuscript effectively correlates smoking with miR-1 dysregulation, it does not establish a detailed mechanistic pathway by which CS alters miR-1 levels specifically in endothelial cells. An in-depth exploration of potential signaling mechanisms or intermediate molecular factors would add to the translational significance of these findings, particularly for targeted therapeutic approaches.

-We greatly appreciate the reviewer's keen insight and suggestions. We have performed a set of experiments to further elucidate the signaling pathway linking CS and miR-1 regulation. (Figure 4H) Since our data showed that VEGFR2 activation is the main axis downstream of CS, we explored the well-known VEGF downstream mediators and found that PI3Kinase/AKT is the main pathway mediating this regulation. These findings confirm our data on the correlation between PI3kinase downstream mediators and miR-1 in the tumor-adjacent lung tissue.(Figure 2D-H) and are also in line with our previous findings on the mechanism of VEGF-induce miR-1 downregulation.(3) we have also now delineated that miR-1 downregulation specifically occurs through degradation of the mature miRNA through a kinetic analysis on endothelial cells. These findings elucidate a novel pathway for miRNA regulation in CS-stimulated ECs and provide several new potential targets for development of anti-cancer therapeutics.

2. While the authors identify several miR-1 target genes associated with tumor progression, the functional relationship between these genes and NSCLC pathophysiology is not fully elucidated. Expanding on how miR-1's regulation of these targets directly influences key cancer processes, such as invasion or metastasis in NSCLC, would enhance the impact of the research. A mechanistic assessment of the

impact of changes in miR-1 or select targets on “field cancerization” or NSCLC characteristics that would pose a worse prognosis would strengthen the report.

-We agree with this point and have made several modifications accordingly. We have clarified the importance of angiogenesis as a basic mechanism of tumor progression (3, 9) and our previous findings on the correlation between miR-1 and overall survival.(3) We have also expanded the discussion on the clinical findings on the role of miR-1 targets in tumors and specifically tumor progression. Our findings on the role of NOTCH3 in angiogenesis also bolsters this significance. (Figure 8F and G)

3. The manuscript does not address potential variations in miR-1 regulation across different endothelial cell subtypes within the lung, which could lead to differential impact on angiogenesis. Does inhibition of miR-1 (by CS) in microvascular lung endothelial cells cause upregulation of similar targets as in NSCLC-derived endothelial cells or pulmonary artery endothelial cells. Investigating this heterogeneity could add valuable context and specificity to the study’s conclusions regarding miR-1’s role in the cancerization field vs. the tumor itself vs. non-malignant detrimental effects of CS on the lung.

-We agree with the point raised by the reviewer. CS may have differential effects on different vascular beds. We tested the effect of CS on three ex vivo cultured Ecs, HUVECs, human lung microvascular ECs and human TECs, and one cell line, EAhy.926. We have thus focused mostly on the microvascular ECs mimicking TECs with angiogenic potential, since these vessels are the most likely site of the angiogenic switch. We do not currently have access to other cell types eg large vein or artery ECs and isolating and characterizing these cells, will entail detailed selection methods that is beyond our current capabilities. However, we have added this point to our section on limitation in the discussion.